# High-intensity training induces non-stoichiometric changes in the mitochondrial proteome of human skeletal muscle without reorganisation of respiratory chain content

Cesare Granata [1,2,9,11✉], Nikeisha J. Caruana [1,3,11], Javier Botella [1], Nicholas A. Jamnick[1,4], Kevin Huynh [5], Jujiao Kuang [1], Hans A. Janssen[1], Boris Reljic [3,10], Natalie A. Mellett[5], Adrienne Laskowski[2], Tegan L. Stait[6], Ann E. Frazier [6,7], Melinda T. Coughlan [2,5], Peter J. Meikle [5], David R. Thorburn [6,7,8], David A. Stroud [3,6,12✉] & David J. Bishop [1,12✉]

Mitochondrial defects are implicated in multiple diseases and aging. Exercise training is an accessible, inexpensive therapeutic intervention that can improve mitochondrial bioenergetics and quality of life. By combining multiple omics techniques with biochemical and in silico normalisation, we removed the bias arising from the training-induced increase in mitochondrial content to unearth an intricate and previously undemonstrated network of differentially prioritised mitochondrial adaptations. We show that changes in hundreds of transcripts, proteins, and lipids are not stoichiometrically linked to the overall increase in mitochondrial content. Our findings suggest enhancing electron flow to oxidative phosphorylation (OXPHOS) is more important to improve ATP generation than increasing the abundance of the OXPHOS machinery, and do not support the hypothesis that training-induced supercomplex formation enhances mitochondrial bioenergetics. Our study provides an analytical approach allowing unbiased and in-depth investigations of training-induced mitochondrial adaptations, challenging our current understanding, and calling for careful reinterpretation of previous findings.

[1] Institute for Health and Sport (iHeS), Victoria University, Melbourne, VIC 3011, Australia. [2] Department of Diabetes, Central Clinical School, Monash University, Melbourne, VIC 3004, Australia. [3] Department of Biochemistry and Pharmacology and Bio21 Molecular Science and Biotechnology Institute, The University of Melbourne, Parkville, VIC 3010, Australia. [4] Metabolic Research Unit, School of Medicine and Institute for Mental and Physical Health and Clinical Translation (iMPACT), Deakin University, Geelong, VIC, Australia. [5] Baker Heart & Diabetes Institute, Melbourne, VIC 3004, Australia. [6] Murdoch Children's Research Institute, Royal Children's Hospital, Melbourne, VIC 3052, Australia. [7] Department of Paediatrics, The University of Melbourne, Melbourne, VIC 3052, Australia. [8] Victorian Clinical Genetics Services, Royal Children's Hospital, Melbourne, VIC 3052, Australia. [9] Present address: Institute for Clinical Diabetology, German Diabetes Center, Leibniz Center for Diabetes Research, Heinrich Heine University, 40225 Düsseldorf, Germany. [10] Present address: Department of Biochemistry and Molecular Biology, Monash Biomedicine Discovery Institute, Monash University, 3800 Melbourne, Australia. [11] These authors contributed equally: Cesare Granata, Nikeisha J. Caruana. [12] These authors jointly supervised this work: David A. Stroud, David J. Bishop. ✉email: cesare.granata@monash.edu; david.stroud@unimelb.edu.au; david.bishop@vu.edu.au

Mitochondria are the main site of energy conversion in the cell and have critical roles in other essential biological processes[1]. Owing to this, defects in mitochondria have been implicated in multiple diseases, medical conditions, and aging[2–5]. The development of interventions to improve the content and function of mitochondria is therefore important to enhance quality of life and to extend life expectancy.

Exercise training is one of the most widely accessible interventions or "therapies" to stimulate mitochondrial biogenesis[6,7]. As arguably the most extensive and also natural perturbation, exercise represents an excellent experimental model for understanding both the complex nature of mitochondrial biogenesis and the plasticity of the skeletal muscle mitochondrial proteome in humans[8]. It is well accepted that exercise training induces an increase in mitochondrial content and respiratory function in human skeletal muscle[9]. However, our knowledge of mitochondrial adaptations is limited to a small fraction of the mitochondrial proteome[10], and is complicated by large training-induced changes in mitochondrial content, which limit the ability to interpret adaptations in individual proteins independent of the overall change in mitochondrial content. To account for this, most studies to date have either performed analyses in mitochondrial isolates obtained after differential centrifugation or have employed a crude normalisation strategy involving the use of markers of mitochondrial content, such as citrate synthase (CS) activity[9,11]. Nonetheless, due to limitations in both approaches[11,12], there remain major gaps in our understanding of the magnitude, timing, and direction of change in individual mitochondrial proteins and related functional pathways, independent of the overall change in mitochondrial content.

A key biological function of mitochondria is energy conversion via oxidative phosphorylation (OXPHOS), which is carried out by the four multi-protein complexes (complexes I to IV [CI-CIV]) of the electron transport chain (ETC) and $F_0F_1$-ATP synthase (or CV)[13]. Assembly of the OXPHOS complexes is an intricate process requiring the coordination of two genomes (nuclear and mitochondrial) and non-subunit proteins known as assembly factors[14]. Only two studies have investigated training-induced changes in OXPHOS complexes in humans[15,16], reporting that exercise differentially modulates the assembly and composition of respiratory chain complexes - including their high molecular weight supercomplex (SC) assemblies. Furthermore, no study has investigated if training-induced changes in chaperones, assembly factors, and single OXPHOS subunits occur in conjunction with, or precede, changes in the complete, functional complexes. Many other processes support assembly and maintenance of the OXPHOS complexes, including mitochondrial protein import and assembly pathways, mitochondrial DNA transcription and translation, metabolite carriers, and pathways producing critical cofactors such as coenzyme Q and iron-sulphur proteins (Fe–S)[14]. In turn, these processes are reliant on pathways supporting mitochondrial biogenesis more broadly, such as lipid biogenesis and maintenance of mitochondrial morphology. Whether these processes and pathways are activated by exercise training and at which specific times remains unknown.

Here, we employed a 3-phase training intervention, combined with multiple omics techniques (transcriptomics, proteomics, and lipidomics) and a normalisation strategy that removed the bias induced by large alterations in mitochondrial content, to investigate mitochondrial adaptations in human skeletal muscle. Our findings revealed an intricate and timely remodelling of the mitochondrial transcriptome, proteome, and lipidome, independent of changes in overall mitochondrial content. Post-normalisation, we identified 185 differentially expressed mitochondrial proteins, representing multiple protein functional classes, as well as bioenergetic and metabolic pathways, that followed distinct patterns of adaptation. In addition, we provide evidence that training-induced changes in the abundance and/or organisation of SCs did not contribute to improvements in mitochondrial bioenergetics. Our study provides an analytical approach that expands our fundamental understanding of training-induced mitochondrial biogenesis. These findings challenge our current knowledge and call for careful reinterpretation of previous results, while also supporting the development of revised hypotheses on how training alters mitochondrial protein content and function, and provide an important resource with implications for both health and disease.

## Results and discussion

**Increases in mitochondrial content underlie changes in markers of mitochondrial biogenesis and mitochondrial respiratory function following training.** The traditional view of mitochondrial biogenesis proposes that changes in individual mitochondrial proteins are stoichiometrically linked to changes in mitochondrial content. Although this has been questioned previously[17], it remains an unresolved matter. To challenge this paradigm, we employed an experimental model in which 10 men were sequentially exposed to different volumes (normal- [NVT], high- [HVT] and reduced- [RVT] volume training) of high-intensity interval training (HIIT) with biopsies taken at baseline (BL) and post-NVT (PN), post-HVT (PH), and post-RVT (PR) (Fig. 1a). This design was chosen as we have previously demonstrated that changes (increases or decreases) in training volume lead to changes in mitochondrial content and markers of mitochondrial biogenesis[9,18–20]. Weekly training volumes for the three phases are shown in Fig. 1b; changes in physiological and performance parameters are presented in Table 1.

We first verified that changes in training volume induced changes in the yield of mitochondrial proteins isolated from muscle biopsies (Fig. 1c), and citrate synthase (CS) activity (Fig. 1d) - a commonly used biomarker of mitochondrial content[21], as previously reported[6,9,20,22–24]. We then confirmed that classic markers of mitochondrial biogenesis, such as the protein content of OXPHOS subunits assessed in whole-muscle lysates (Supplementary Fig. 1a, upper panels), as well as measurements of mitochondrial respiratory function, such as mitochondrial respiration in permeabilised fibres and ETC enzyme activity (Supplementary Fig. 1b, c, respectively, all upper panels) changed, for the most part, as a consequence of changes in training volume. RVT resulted in no change in any of these markers, suggesting that the training stimulus was sufficient to prevent a decrease in these parameters. Subsequently, to investigate the underpinnings of training-induced changes in the above markers, we normalised these values by CS activity - a widely used normalisation strategy to account for differences in mitochondrial content[9,20,21,25]. This demonstrated that, for the most part, absolute changes in markers of mitochondrial biogenesis and respiratory function were stoichiometrically associated with the overall increase in mitochondrial content (Supplementary Fig. 1a, b, c, all lower panels). This is consistent with previous findings[20] and with the notion that absolute changes in markers of mitochondrial respiratory function (i.e., mitochondrial respiration and mitochondrial enzyme activity) induced by manipulation of training volume are mainly driven by the overall change in mitochondrial content, rather than by altered mitochondrial efficiency per se[9,20,26].

**Disentangling changes in the mitochondrial proteome from the general increase in mitochondrial content observed post-training.** The above findings seem to indicate that altering training volume results in a stoichiometric relationship between

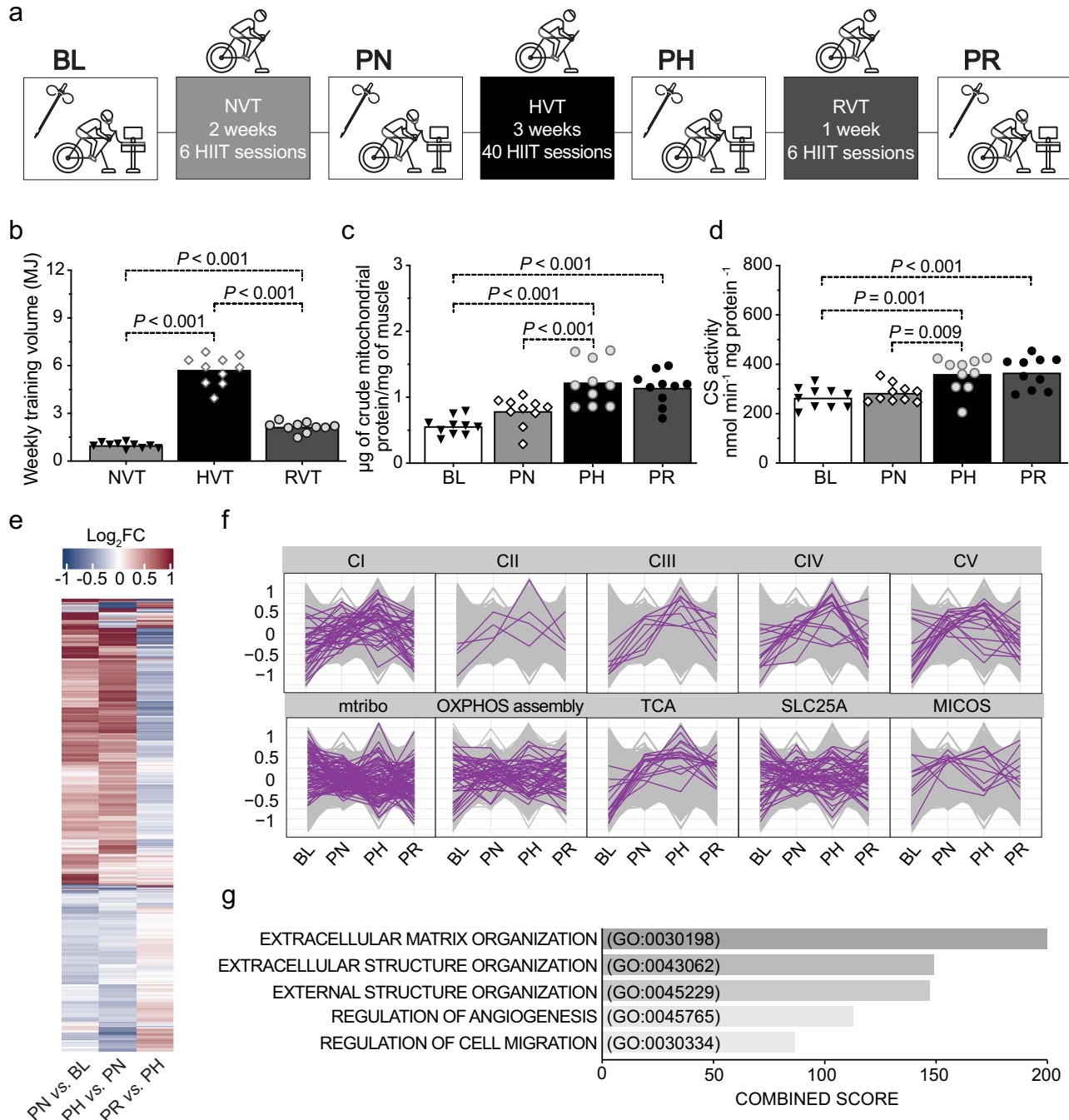

training-induced changes in mitochondrial content and markers of mitochondrial biogenesis and mitochondrial respiratory function[9]. However, whether there is also a fixed stoichiometry between changes in mitochondrial content and individual mitochondrial proteins, protein functional classes, and metabolic pathways, remains undetermined. To gain further insight into the effects of training on mitochondria we first employed RNA sequencing (RNA-seq) based transcriptomics (Supplementary Data 1). We identified 1206 transcripts differentially expressed across the three training phases (Fig. 1e and Supplementary Data 2), and we observed the expected increase in gene transcripts encoding subunits of the OXPHOS complexes (Fig. 1f, upper panels, compare with Supplementary Fig. 1a, upper panels) and enzymes of the tricarboxylic acid (TCA) cycle (Fig. 1f, central lower panel), consistent with previous studies in humans[27].

However, transcripts for genes encoding other mitochondrial proteins did not all follow the same trend (Fig. 1f, lower panels). Moreover, gene ontology enrichment analysis of the differentially expressed transcripts did not identify significant alterations of pathways involved in mitochondrial respiratory function (Fig. 1g and Supplementary Data 2), despite the increase in mitochondrial respiration and ETC enzyme activity reported above (Supplementary Fig. 1b, c, respectively). This reveals a greater complexity in the transcriptional responses to exercise than previously hypothesised; i.e., not all mitochondria-related gene transcripts changed in the same direction and with a similar magnitude following different training phases (Fig. 1f).

As transcriptomics suggested there may not be a fixed stoichiometry between training-induced changes in mitochondrial content and individual mitochondrial proteins, we sought to

**Fig. 1 A complex network of transcriptional responses underpins training-induced increases in mitochondrial content. a** Study design; coloured boxes indicate a training phase: normal-volume training (NVT), high-volume training (HVT), and reduced-volume training (RVT), whereas open boxes indicate a testing session and resting skeletal muscle biopsy: baseline (BL), post-NVT (PN), post-HVT (PH), and post-RVT (PR). **b** Weekly training volume during the NVT, HVT, and RVT training phase; training volume was calculated by multiplying the absolute exercise intensity in Watts by the effective duration of exercise training in minutes (excluding the warm up and the rest periods between intervals) by the total number of training sessions in each phase. **c** Mitochondrial yield per mg of tissue achieved during mitochondria isolation of human vastus lateralis biopsy samples (biochemical enrichment of mitochondrial protein). **d** Citrate synthase (CS) enzyme activity assessed in whole-tissue (vastus lateralis) homogenates. For panel **b**–**d** source data are provided as a Source Data file and $P$ values are indicated on the figure to three decimal places; for $P$ values that were truncated, the corresponding accurate $P$ values were: HVT vs. NVT: $P = 3.9e^{-14}$; RVT vs. NVT: $P = 3.8e^{-5}$; HVT vs. NVT: $P = 1.1e^{-12}$ in panel **b**; PH vs. BL: $P = 1.9e^{-6}$; PR vs. BL: $P = 1.6e^{-5}$; PH vs. PN: $P = 8.3e^{-4}$ in panel **c**; and PR vs. BL: $P = 5.3e^{-4}$ in panel **d**. **e** Heatmap of differentially expressed transcripts between training phases determined with an adjusted $P < 0.05$ (Benjamini–Hochberg). Row clustering determined by unsupervised hierarchical cluster analysis. **f** Profile plots showing relative scaled transcripts of subunits of the five oxidative phosphorylation (OXPHOS) complexes (CI to CV), mitochondrial ribosomes (mtribo), OXPHOS assembly, TCA cycle, SLC25As, and mitochondrial contact site and cristae organising system (MICOS) complex. Transcripts grouped according to known literature (see Methods section). **g** Biological process (BP) gene ontology of all differentially expressed transcripts as in **e**; the top five biological processes by adjusted $p$ value, as determined by *Enrichr* (see Methods section), are displayed with their combined score (the full list is presented in Supplementary Data 2). HIIT high-intensity interval training, MJ megajoules, CI-V complex I–V, IB immunoblotting. Datasets **b**–**d** ($n = 10$) were analysed by repeated measures one-way ANOVA followed by Tukey's post hoc testing; $P < 0.05$. Filled triangles, empty diamonds, and empty and filled circles represent individual values; the maxima of each bar represents the mean value. Datasets **e**–**g**, $n = 5$.

---

**Table 1 Participants' physiological and endurance performance measurements.**

| Measurement | BL | PN | PH | PR |
|---|---|---|---|---|
| Age | 22.3 ± 5.2 | – | – | – |
| Height | 178.1 ± 10.7 | – | – | – |
| Body mass | 78.8 ± 12.1 | 79.0 ± 12.9 | 78.0 ± 12.5 | 77.9 ± 11.8# |
| BMI | 24.8 ± 2.9 | 24.9 ± 3.2 | 24.6 ± 3.0 | 24.5 ± 2.9# |
| $\dot{V}O_{2Peak}$ | 46.7 ± 8.2 | 47.7 ± 8.8 | 50.5 ± 6.0*# | 50.0 ± 6.0* |
| $\dot{W}_{LT}$ | 174.1 ± 30.7 | 182.5 ± 33.0 | 222.2 ± 36.2*# | 219.6 ± 31.4*# |
| $\dot{W}_{peak}$ | 231.1 ± 43.2 | 233.2 ± 43.5 | 266.3 ± 42.4*# | 264.5 ± 40.4*# |
| 20k-TT time | 2321.0 ± 168.1 | 2309.1 ± 257.4 | 2094.6 ± 130.8*# | 2030.1 ± 140.5*# |

*BL* baseline, *PN* post normal-volume training, *PH* post high-volume training, *PR* post reduced-volume training, *BMI* body mass index [kg m$^{-2}$], $\dot{V}O_{2Peak}$ peak oxygen uptake [mL min$^{-1}$ kg$^{-1}$], $\dot{W}_{LT}$ power at the lactate threshold [Watts], $\dot{W}_{peak}$ peak power output [Watts], *20k-TT* 20-km time trial [seconds].
Source data are provided as a Source Data file. $P$ values reported in the table were: $P = 0.040$ for PR vs. PN in "Body mass"; $P = 0.041$ for PR vs. PN in "BMI"; $P = 0.004$ for PH vs. BL, $P = 0.046$ for PH vs. PN, and $P = 0.014$ for PR vs. BL in "$\dot{V}O_{2Peak}$"; $P = 4.5e^{-10}$ for PH vs. BL, $P = 2.5e^{-8}$ for PH vs. PN, $P = 1.5e^{-9}$ for PR vs. BL, and $P = 9.6e^{-8}$ for PR vs. PN in "$\dot{W}_{LT}$"; $P = 1.1e^{-8}$ for PH vs. BL, $P = 3.8e^{-8}$ for PH vs. PN, $P = 3.3e^{-8}$ for PR vs. BL, and $P = 1.1e^{-7}$ for PR vs. PN in "$\dot{W}_{peak}$"; $P = 4.3e^{-6}$ for PH vs. BL, $P = 1.0e^{-5}$ for PH vs. PN, $P = 4.6e^{-8}$ for PR vs. BL, and $P = 1.0e^{-7}$ for PR vs. PN in "20k-TT time". All values are mean ± SD; $n = 10$ for all analyses; all datasets analysed by repeated measures one-way ANOVA followed by Tukey's post hoc testing; $P < 0.05$ * vs. BL, # vs. PN.

---

support these findings with quantitative proteomics. To negate the general increase in mitochondrial content following exercise training (Fig. 1d) and to avoid this inherent bias on our proteomics measurements, we performed label-free quantitative (LFQ) proteomics on equal amounts of mitochondria-enriched fractions (biochemical enrichment of mitochondrial proteins using differential centrifugation). We quantified 1411 proteins — 726 of which were annotated as mitochondrial based on either the Mitocarta2.0[28] or the Integrated Mitochondrial Protein Index (IMPI; Known and Predicted Mitochondrial)[29] databases (Supplementary Data 3). Considering only the high-confidence IMPI (Known Mitochondrial) dataset, mitochondrial proteins represented 41% of the total number of proteins identified in our mitochondria-enriched fraction (584 annotated mitochondrial proteins), contributing to 32% of the overall protein abundance based on raw intensity data for each protein (Supplementary Data 3). This demonstrated the presence of a significant proportion of co-isolating non-mitochondrial proteins, with enrichment analysis indicating these to be predominantly of myofibrillar origin (Supplementary Fig. 2a), consistent with most mitochondria in skeletal muscle being tightly associated with myofibrils[23]. Surprisingly, we still observed a relative increase in the abundance of mitochondrial proteins within mitochondrial isolates following training (Fig. 2a, upper panel, magenta profiles), an artefact that likely stems from differences in mitochondrial protein enrichment (MPE) - the contribution of mitochondrial protein intensities in biochemically isolated

mitochondria relative to all protein intensities - at different time points. Indeed, the MPE for the four different time points in the present study was significantly different (Fig. 2a, middle panel) and corresponded to 24.5% at baseline (BL), 31.5% post-NVT (PN), 39.3% post-HVT (PH), and 34.5% post-RVT (PR) (Supplementary Data 3) - a pattern consistent with our assessment of mitochondrial content (compare Fig. 2a, middle panel with Fig. 1c, d).

The above findings highlight that the standard approach of relying on biochemical enrichment to eliminate the bias arising from different MPEs across time points is insufficient and that an additional strategy is needed. We therefore complemented our biochemical enrichment approach with a statistical correction strategy consisting of (i) removing proteins that were identified in less than 70% of samples, (ii) removing all except high-confidence mitochondrial proteins (IMPI Known Mitochondrial), and (iii) undertaking variance stabilising normalisation (VSN) (see Methods section for more detail). This resulted in the retention of 498 mitochondrial proteins normalised across the dataset with high confidence (Supplementary Data 4). Unlike CS activity, which is a single value for each sample, our normalisation strategy utilises the trends of hundreds of proteins and reduces the confounding effect of potential outliers. This is extremely important, given that previous studies have shown a greater than 5-fold difference in training-induced synthesis rate between fast and slow synthesised mitochondrial proteins in human skeletal muscle[11,30,31]. Plotting the resulting data revealed that the general

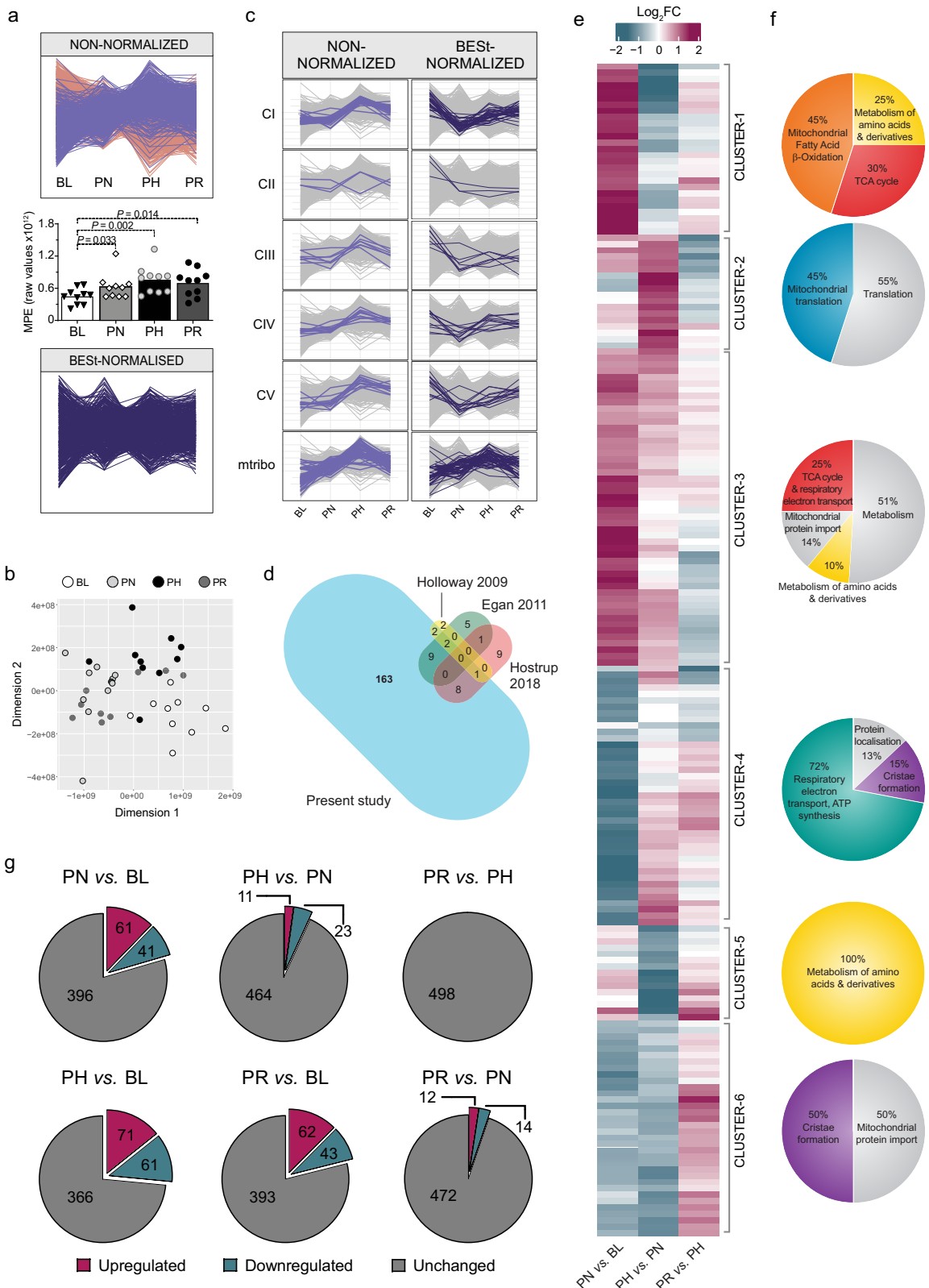

increase in mitochondrial proteins with different training volumes had been compensated for, indicating that our biochemical enrichment and statistical correction (BESt-normalisation) was successful (Fig. 2a, lower panel). In further support of our normalisation strategy, multidimensional scaling analysis showed good segregation of samples corresponding to the different time points (Fig. 2b).

increase in mitochondrial proteins with different training volumes had been compensated for, indicating that our biochemical enrichment and statistical correction (BESt-normalisation) was successful (Fig. 2a, lower panel). In further support of our normalisation strategy, multidimensional scaling analysis showed good segregation of samples corresponding to the different time points (Fig. 2b).

To assess the power of our normalisation strategy, we compared results from non-normalised and BESt-normalised proteomics values. Whereas analysis of non-normalised intensity values revealed a lack of change in the abundance of OXPHOS and mitochondrial ribosomal subunits post-NVT, followed by a marked increase post-HVT, and a decrease post-RVT (Fig. 2c, left panels), results from BESt-normalised values showcased a very

**Fig. 2 Correcting for changes in mitochondrial content reveals differentially expressed protein clusters with divergent prioritisation in response to training. a** Training-induced differences in the mitochondrial protein content of isolated mitochondrial (IM) fractions obtained from vastus lateralis muscle biopsies. Upper panel: profile plot of all non-normalised intensity values displaying "known mitochondrial" (IMPI "Known Mitochondrial") (magenta) and non-mitochondrial (salmon) proteins co-precipitating during mitochondrial isolation. Proteins identified in less than 70% of samples were removed. Middle panel: mitochondrial protein enrichment (MPE) obtained by adding the raw intensity values from LC-MS/MS analysis of all "known mitochondrial" proteins (IMPI "Known Mitochondrial") at the four different time points. Filled triangles, empty diamonds, and empty and filled circles represent individual values; the maxima of each bar represents the mean value; analysed by repeated measures one-way ANOVA followed by Tukey's post hoc testing; $P < 0.05$ ($n = 10$ for all groups). Lower panel: same as upper panel but following removal of non-mitochondrial proteins and statistical correction (BESt-) normalisation. Source data are provided as a Source Data file. **b** Multidimensional scaling analysis showing segregation of samples obtained from IM fractions of BESt-normalised proteomics values. **c** Scaled profile plots showing the relative abundance of subunits of OXPHOS complexes and mtribo of the non-normalised intensity values of "known mitochondrial" proteins (IMPI "Known Mitochondrial") following valid value filtering (left panels), and values obtained after BESt-normalisation (right panels). **d** Venn diagram representing the overlap in differentially expressed proteins between the present study and previous literature utilising young healthy humans[23,32,33]. The bubble size for each study is proportional to the number of differentially expressed proteins identified (this was not possible for the overlapping size). **e** Heatmap of BESt-normalised differentially expressed mitochondrial proteins (IMPI "Known Mitochondrial"), identified with an adjusted $P < 0.01$ (Benjamini–Hochberg) determined by linear model fit using *limma* with empirical Bayes method. Row clustering determined by unsupervised hierarchical cluster analysis. **f** Venn diagram representation of the gene ontology of the six clusters identified in e obtained from enrichment analysis (using the ClueGO [v2.5.6] application in Cytoscape [v3.7.1]; see Methods section) of all proteins belonging to each cluster performed separately (Supplementary Data 5). **g** Venn diagram representations of the number of BESt-normalised differentially expressed mitochondrial proteins (IMPI "Known Mitochondrial") between each of the time points. Differentially expressed proteins for each time point comparison were identified by linear modelling with multiple comparison adjustment made with a $P$ value of <0.01 using the Benjamini–Hochberg method, as shown in Supplementary Data 7. BL baseline, PN post-NVT, PH post-HVT, PR post-RVT. $n = 10$ for all analyses.

different picture, as the content of OXPHOS subunits was markedly decreased (deprioritised) post-NVT, and showed discreet and specific changes (mainly modest increases) thereafter (Fig. 2c, right panels; results discussed in more detail later), whereas mitochondrial ribosomal subunits increased to a greater extent than mitochondrial content (prioritised) (Fig. 2c, right panels). These findings demonstrate the power of our BESt-normalisation strategy compared to classical approaches, as it enables an unbiased comparison of the changes in individual mitochondrial proteins relative to the overall increase in mitochondrial content following training, providing clear evidence that these changes are not stoichiometrically linked. As a means of comparison, we also employed the classical approach of normalising the abundances of specific proteins derived from SDS-PAGE and immunoblot analysis on isolated mitochondrial fractions by CS activity; this demonstrates that while this widely used approach partially corrects for changes in MPE, it fails to recapitulate most of the findings observed with our BESt-normalisation strategy (Supplementary Fig. 2b).

From this point onwards, unless specified otherwise, post BESt-normalisation proteomics results will be presented, enabling the investigation of training-induced changes in mitochondrial proteins without the confounding influence of changes in mitochondrial content.

**Discovery of mitochondrial proteins and functional classes and metabolic pathways differentially affected by exercise training.** We next interrogated our BESt-normalised mitochondrial proteome to gain new insights into non-stoichiometric changes in individual mitochondrial proteins following three training phases designed to induce different effects on mitochondrial biogenesis. Using linear modelling (comparable to a multiple samples ANOVA), we identified 185 mitochondrial proteins that were differentially expressed across the three training phases (Supplementary Data 5). This is ~10-fold greater than the number of proteins reported to change post-training in previous human skeletal muscle studies[23,32–34] (Fig. 2d). This increased number of proteins was likely related to our normalisation strategy, and did not seem to depend on the large training volume utilised (120 proteins were differentially expressed after the modest training volume of the NVT phase alone), nor was the result of comparing multiple time points (118 differentially expressed proteins would

have been identified by sampling muscle only at the beginning and end of this study).

Unsupervised hierarchical clustering of all differentially expressed mitochondrial proteins revealed six clusters (Fig. 2e) with distinct patterns of change in response to training (Supplementary Fig. 2c), which also confirmed the absence of a fixed stoichiometry between training-induced changes in mitochondrial content and mitochondrial protein functional classes. Enrichment analysis of differentially expressed proteins within each of the six clusters revealed them to be enriched in, but not always exclusively contain, proteins involved in fatty acid β-oxidation (FAO), mitochondrial translation, the TCA cycle, OXPHOS, metabolism of amino acids, and cristae formation, respectively (Fig. 2f and Supplementary Data 5). Because not all proteins belonging to each individual Reactome pathway segregated within one cluster (Supplementary Fig. 2d), we compared our 185 differentially expressed proteins with proteins belonging to the six aforementioned functional classes (Reactome R-HSA) and other related pathways; matching proteins (Supplementary Data 6) were used to generate the individual pathway heatmaps presented in Fig. 3 and to discuss our findings. In addition, because Reactome, like other protein databases, only provides an extensive but not exhaustive list of proteins involved in each pathway, we complemented our heatmaps with other proteins identified through literature (see Methods section for more information on how pathways in Supplementary Data 6 and Fig. 3 were generated).

To pinpoint the origin of training-induced changes in individual mitochondrial proteins, we performed unpaired *t*-tests between group pairs following permutation-based false discovery rate (FDR) correction. Using a conservative approach, we matched the differentially expressed proteins identified by each *t*-test comparison with the 185 differentially expressed proteins across the three training phases; this analysis, which includes all relevant comparisons, including those between each time point against baseline, can be found in Supplementary Data 7. The majority of changes occurred during the initial NVT phase (Fig. 2g and Supplementary Data 7), where 102 proteins were differentially expressed, suggesting that the greatest number of mitochondrial adaptations took place within the first few training sessions and these became harder to obtain as the training progressed (Fig. 2e, note the strongest changes are found in the

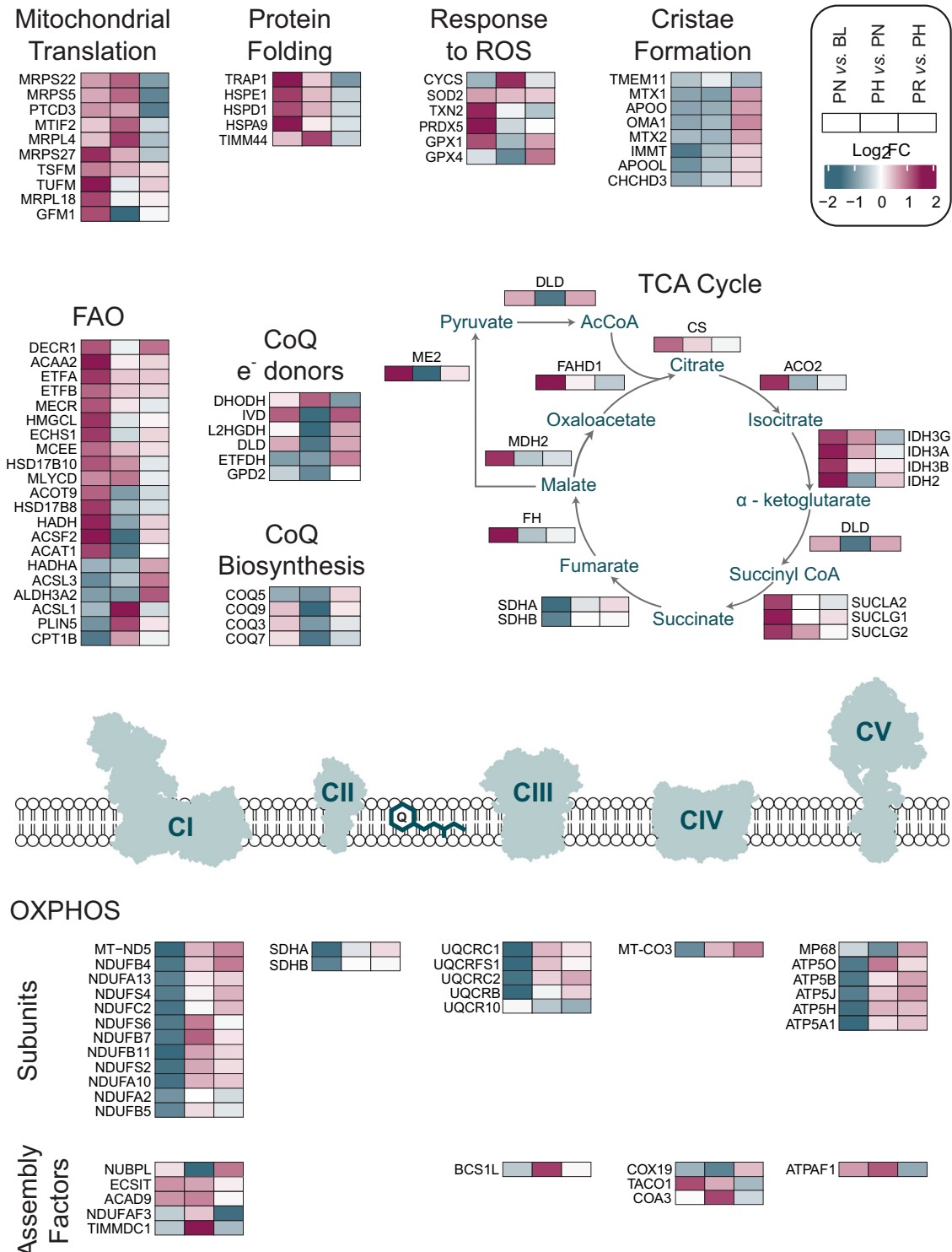

**Fig. 3 Overview of training-induced changes in mitochondrial protein functional classes and metabolic pathways.** Protein functional classes and/or metabolic pathways (defined in Supplementary Data 6) as determined in Supplementary Data 7. Row clustering determined by unsupervised hierarchical cluster analysis. BL baseline, PN post-NVT, PH post-HVT, PR post-RVT, ROS reactive oxygen species, FAO fatty acid β-oxidation, CoQ Coenzyme Q, TCA tricarboxylic acid cycle, OXPHOS oxidative phosphorylation.

"PN *vs*. BL" column). Our study also confirms that the training stimulus during the RVT phase was sufficient to prevent a loss of adaptation (Fig. 2g and Supplementary Data 7). The lack of change in mitochondrial markers in the current study is in contrast with the significant reductions reported in a previous study[20]; this is likely to stem from the greater training volume and/or the shorter duration of the RVT phase employed in the

present study (2.14 MJ/week for 1 week vs. 0.32 MJ/week for 2 weeks, respectively).

**Proteins important for the translation of mitochondrial proteins are sensitive to training volume.** We next examined the six over-represented pathways highlighted by our enrichment

analysis, beginning with the pathway related to mitochondrial protein translation (Fig. 3), which mostly segregated within cluster-2 (Fig. 2f; Supplementary Fig. 2d; and Supplementary Data 5). Specifically, we identified proteins from both the small 28S (MRPS5, MRPS22, MRPS27) and large 39S (MRPL4, MRPL18) subunits of the mitochondrial ribosome, as well as proteins involved in translation initiation (MTIF2) and elongation (GFM1, PTCD3, TSFM, TUSM; Fig. 3). Whereas only four proteins were upregulated post-NVT, eight of the 10 aforementioned proteins were upregulated post-HVT, with six remaining elevated post-RVT (Supplementary Data 7). These findings demonstrate that training-induced changes occur in these or related mitochondrial translation proteins in young, healthy humans, in line with observations previously made in older (65–80 years) individuals[34]. Although the influence of training duration cannot be completely ruled out, our findings also indicate that changes in proteins involved in mitochondrial protein translation may be sensitive to training volume, consistent with a study observing a positive association between these proteins and physical activity levels in healthy individuals (20–87 years)[35].

**Mitochondria prioritise the TCA cycle and FAO in response to NVT.** Proteins involved in the TCA cycle pathway grouped mainly within cluster-3 and, to a lesser extent, cluster-1 (Fig. 2f; Supplementary Fig. 2d and Supplementary Data 5). This pathway was characterised by a large and concerted increase post-NVT (12 of 16 enzymes were significantly increased), with many of these proteins remaining upregulated compared to BL both post-HVT and post-RVT (Fig. 3 and Supplementary Data 7). Of note, SDHA and SDHB, subunits of CII - the only TCA cycle enzyme that also participates in the ETC - clustered with the majority of OXPHOS subunits (discussed below) and decreased during NVT. While training-induced changes in some TCA cycle proteins have been previously reported[23,33], our data extends these findings by revealing training-induced increases in ACO2, SUCLG1, SUCLG2, SUCLA2, and FAHD1, noting that that these enzymes adapted mostly in a coordinated fashion.

Proteins involved in FAO segregated mainly within cluster-1 and cluster-3 (Fig. 2f; Supplementary Fig. 2d and Supplementary Data 5). Our data demonstrated a large and concerted increase in these proteins post-NVT, with minimal (HVT) and no (RVT) changes thereafter (Fig. 3 and Supplementary Data 7). Proteins increased post-NVT included chain shortening enzymes involved in FAO (ACAA2, ACSF2, ECHS1, HADH, HSD17B10, MECR, MLYCD) and enzymes required to convert unsaturated fatty acids into intermediates of FAO (DECR1 and HSD17B8). Similar increases were also observed for ETFA and ETFB, which transfer electrons produced during FAO to the OXPHOS system, further highlighting the early increase in FAO. HMGCL and ACAT1, two enzymes involved in the ketogenesis pathway, were also increased, suggesting that excess acetyl-CoA generated by enhanced FAO may be fed into ketogenesis. Taken together, these findings indicate that the synthesis of proteins involved in both the TCA cycle and FAO pathways is prioritised in the early stages of an endurance training intervention; these increases may have contributed, at least in part, to the training-induced increase in mitochondrial respiration (Supplementary Fig. 1b, upper panels).

**Lipidomics highlights a divergent response to training of different lipid classes.** To highlight specific changes in fatty acid composition, LC-MS/MS based comparative lipidomics was performed using the same mitochondrial isolates used for proteomics. A total of 779 species representing 30 lipid classes (Supplementary Data 8) were quantified. Indeed, changes in the mitochondria-specific cardiolipins (CLs), the biomarker most strongly associated with mitochondrial content in human skeletal muscle[21], mirrored changes in training volume and in non-normalised mitochondrial proteins (compare Fig. 4a, left panel with Fig. 2a, upper panel; Supplementary Data 8). Therefore, to ensure that our lipidomics data were corrected for changes in MPE in our fractions, we applied a similar approach to the one used for proteomics where lipidomic results were normalised according to the trend observed for the CL class (see Methods section). Post-normalisation profile plots of CL species indicated that, despite changes in individual CL species, the general increase in mitochondrial content with different training volumes had been compensated for by our normalisation strategy (Fig. 4a; right panel), and the lipid profiles could be used to readily segregate samples across different time points (Fig. 4b). From this point on we will present post-normalisation lipidomics results, unless specified otherwise.

We identified 182 differentially expressed lipid species (Fig. 4c and Supplementary Data 9). Similar to what we observed in our proteomics dataset, and consistent with previous research in humans[36,37], our study highlighted the divergent response to training of different lipid classes and species relative to overall changes in mitochondrial content (Fig. 4d). However, in contrast to proteomics, where the greatest number of changes were observed early (during NVT) and no changes were observed during RVT, our lipidomics assessment showed the most significant changes to occur during the latter two stages (Supplementary Fig. 3). This suggests a delayed response to changes in lipid content compared to proteins, and may reflect overall mitochondrial biogenesis as training-induced changes in lipid species were mainly related to mitochondrial membrane-based lipids (CLs, phosphatidylethanolamines and phosphatidylcholines) and the triglycerides (TGs) (Supplementary Fig. 3). Contrary to previous training studies in obese humans[36–38], we did not observe reductions in muscle ceramides [with the exception of Cer(d18:1/18:0) during HVT] or diacylglycerol content in our healthy, young men following training. While this could indicate a different response of lipid species in mitochondria isolates (our study) compared to whole-muscle lysates[36–38], as previously suggested[39], it could also highlight adaptation differences between obese and healthy individuals. A complete dataset detailing the differentially expressed lipids between each of the four time points (including comparisons of all time points vs. baseline) can be found in Supplementary Data 9.

Of relevance to the increase in FAO reported above, we observed a post-NVT decrease in TGs (Fig. 4d), a class of lipids used as an important source of energy production during exercise[40]; however, likely due to the high number of multiple comparisons in our bioinformatic analysis ($n = 779$), no significant changes in single TG species were detected (Supplementary Fig. 3). The decrease in TGs in our study is consistent with an increase in fatty acid turnover, and is corroborated by the post-NVT increase in FAO enzymes (Fig. 3). Finally, changes in TGs (Fig. 4d), which are not found in mitochondria but rather in lipid droplets[41], mirrored changes in PLIN5 (Fig. 3). Since PLIN5 is a protein that tethers mitochondria to lipid droplets and regulates the release of lipids for FAO[42,43], we hypothesise that training-induced changes in TGs may reflect increased association of lipid droplets with mitochondria. While few studies have evaluated the effects of different training interventions on skeletal muscle lipid species, our results add to the growing evidence that changes with training are specific to the exercise prescription and that training-induced changes in individual lipid species and lipid classes are not stoichiometrically linked.

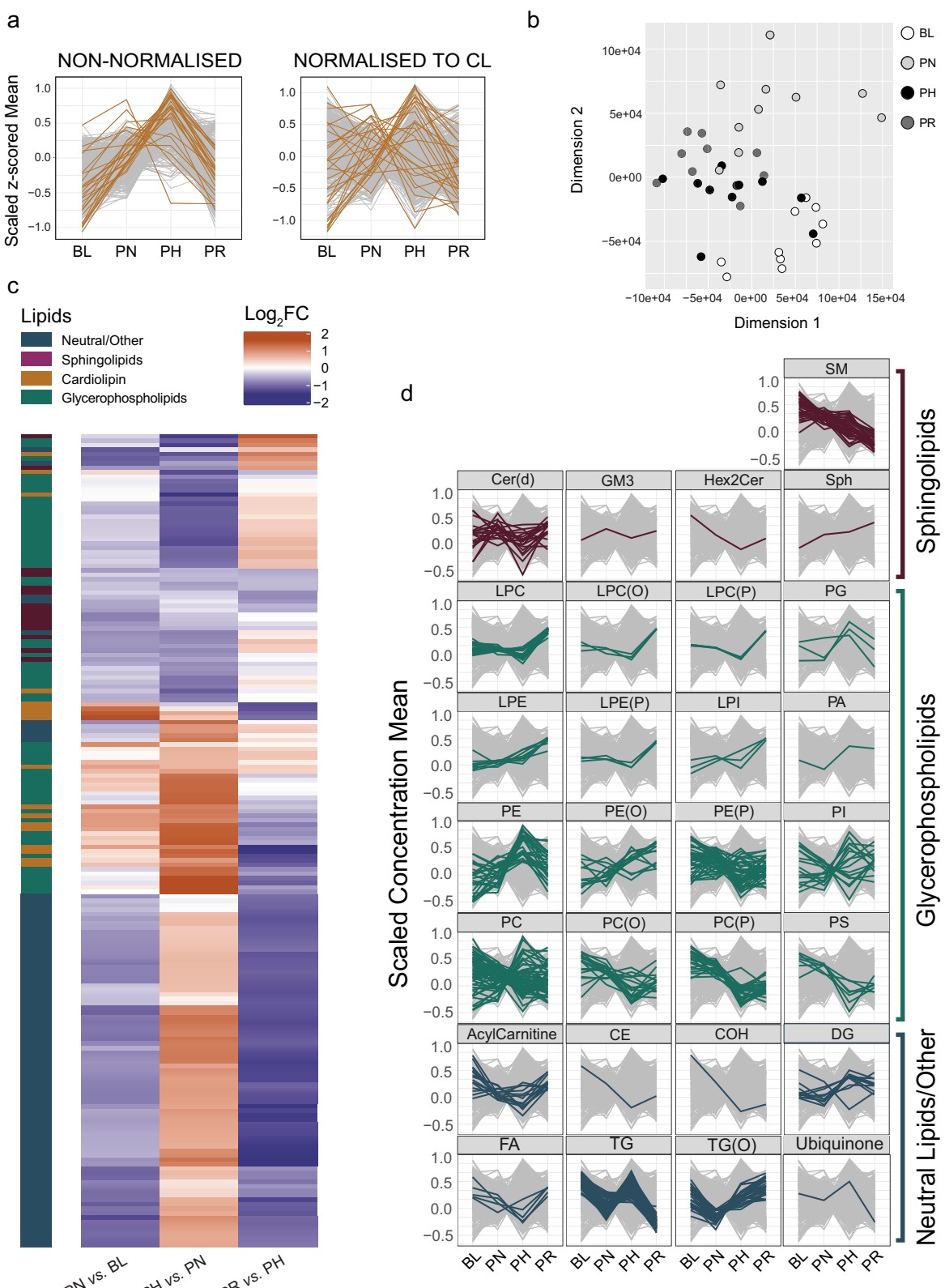

**A training-induced increase in OXPHOS subunits is not a priority to enhance mitochondrial respiration**. We next examined the OXPHOS pathway, which segregated almost entirely in cluster-4 (Fig. 2f; Supplementary Fig. 2d and Supplementary Data 5). Our analysis provided excellent coverage of OXPHOS proteins and identified 78 OXPHOS subunits. We report training-induced changes in 26 OXPHOS subunits (Fig. 3),

equivalent to 33% of all subunits identified. Differentially expressed OXPHOS proteins included subunits from all six of the modules required for the assembly of CI, both catalytic subunits of CII, as well as subunits from CIII, one of the three core subunits of CIV, and proteins found in both functional domains of CV. Specifically, we observed a large and coordinated (23 subunits) decrease post-NVT, with no changes during HVT and/or

**Fig. 4 Lipidomics highlights a divergent response to training of different lipid classes. a** All-lipids profile plot of non-normalised (left panel) and cardiolipin (CL) class-normalised (right panel) intensity values from mitochondrial isolates, displaying all individual CL species (gold) and all other lipid species (grey). **b** Multidimensional scaling analysis showing segregation of samples obtained from IM fractions of lipidomics values normalised by the entire CL class. **c** Heatmap of differentially expressed lipids after normalisation by the entire CL class between training phases determined by linear model fit using *limma* with empirical Bayes method with an adjusted $P < 0.05$ (Benjamini–Hochberg) to account for multiple comparisons. Row clustering determined by unsupervised hierarchical cluster analysis. **d** Profile plots of the scaled concentration mean of different lipid classes. For each profile plot, all the individual lipid species identified within the specific class are represented in colour over the entire lipidome (grey) at each time point. The intensity values for all lipid species were normalised by the entire CL class to adjust for changes in mitochondrial content. BL baseline, PN post-NVT, PH post-HVT, PR post-RVT. $n = 10$ for all analyses. SM sphingomyelins, Cer(d) ceramides, GM3 monosialodihexosylganglioside, Hex2Cer dihexosylceramide, Sph sphingosine, LPC lysophosphatidylcholines, LPC(O) lysoalkylphosphatidylcholines, LPC(P) lysoalkenylphosphatidylcholines, PG phosphatidylglycerols, LPE lysophosphatidylethanolamines, LPE(P) lysoalkenylphosphatidylethanolamines, LPI lysophosphatidylinositols, PA phosphatidic acid, PE phosphatidylethanolamines, PE(O) alkylphosphatidylethanolamines, PE(P) alkenylphosphatidylethanolamines, PI phosphatidylinositols, PC phosphatidylcholines, PC(O) alkylphosphatidylcholines, PC(P) alkenylphosphatidylcholines, PS phosphatidylserines, CE cholesterol ester, COH cholesterol, DG diacylglycerols, FA fatty acids, TG triacylglycerols, TG(O) alkyl triacylglycerols.

RVT (Fig. 3 and Supplementary Data 7). Changes in SLC25A4 (ANT1), a protein catalysing the exchange of cytosolic ADP and mitochondrial ATP across the mitochondrial inner membrane[44], mirrored these changes (Supplementary Data 7). The post-NVT decrease in OXPHOS subunits indicates that during times of increased mitochondrial biogenesis, synthesis of proteins involved in OXPHOS pathways is deprioritised. Moreover, our results demonstrated a large post-NVT increase in TCA cycle and FAO related enzymes - the major providers of reducing equivalents (NADH and FADH$_2$) to the OXPHOS system[45]; this suggests that enhancement of these two metabolic pathways may be more important to improve mitochondrial respiration following exercise training (Supplementary Fig. 1b, upper panels) than an increase in OXPHOS subunits.

An increase in reducing equivalents coupled with the deprioritisation of the OXPHOS machinery could lead to increased reactive oxygen species (ROS) generation. Here we report an immediate post-NVT increase in the abundance of enzymes involved in protection from oxidative stress, such as PRDX5 (the most increased protein), TXN2, and GPX1[46] (Fig. 3 and Supplementary Data 7). Moreover, both SOD2, the mitochondrial ROS scavenger[46], and PRDX5 were increased post-HVT and post-RVT. These findings suggest that training-induced synthesis of proteins involved in the protection from oxidative stress was emphasised early during the training intervention and maintained throughout, consistent with the notion that exercise training provides protection against oxidative stress[46,47].

The coordinated assembly of OXPHOS complexes involves various assembly factors, chaperones, and protein translocation components, and has been suggested to be in the order of days[48]. Whereas increases in OXPHOS assembly factors occurred mostly post-HVT (Fig. 3 and Supplementary Data 7), changes in subunits of the mitochondrial ribosome (Fig. 2c, lower right panel), which synthesise the 13 OXPHOS subunits encoded by mitochondrial DNA, and in several chaperone proteins regulating import and folding of proteins destined for the mitochondrial matrix, took place earlier (post-NVT; Fig. 3 and Supplementary Data 7). Chaperone proteins upregulated post-NVT included HSPA9 (mtHsp70), HSPD1 (mtHsp60), HSPE1 (mtHsp10), and TRAP1 (mtHsp75), whereby TIMM44 was increased during HVT (Fig. 3 and Supplementary Data 7). All five proteins remained elevated compared to BL both post-HVT and post-RVT. Except for HSPE1[23], none of the above chaperones have previously been reported to respond to exercise. Owing to the notion that all of these chaperones are known to interact[49], our data indicate a coordinated increase in the protein quality control system in response to exercise training. While we detected many subunits of the translocases of the outer and inner membrane (TOM and

TIM, respectively; Supplementary Data 3), which are required for importing nuclear-encoded subunits[50], most did not change significantly with training, suggesting that their levels remain stable relative to changes in mitochondrial content.

**Training-induced changes in mitochondrial content, but not changes in the abundance and/or organisation of SCs, mediate improvements in mitochondrial respiration.** A previous study reported that training increases the formation of ETC SCs - high molecular weight assemblies comprised of CI, CIII and CIV[51] - and suggested this to contribute to improvements in mitochondrial respiration[15]. Although SCs were originally proposed to support enhanced ETC function[52], their role has become controversial, with multiple groups reporting that they confer no bioenergetic advantage[53]. To provide further insight, we performed BN-PAGE analysis of ETC SCs (Fig. 5a). We observed increases in SC abundance with increasing training volumes, achieving significance post-HVT (Fig. 5b, upper panels); however, these increases were no longer significant upon normalisation by MPE (Fig. 5b, middle panels).

We next assessed whether exercise training affects the distribution of ETC complexes into SCs. In contrast to the only previous report showing a post-training redistribution of CIII and CIV (but not CI) into SCs in elderly individuals[15], and recent research in mouse skeletal muscle showing redistribution of CIV into SCs[54], our analysis revealed no significant changes throughout the intervention (Fig. 5b, lower panels). This demonstrates that in young healthy men endurance training does not alter the ETC complex distribution within major and minor SC assemblies. While the contrasting findings may relate to either species differences (i.e., mice vs. humans) or differences in the participants' average age (66 vs. 22 y) and baseline $\dot{V}O_{2max}$ (1.9 vs. 3.6 L min$^{-1}$), as well as the training interventions (48 sessions of moderate-intensity training in 16 weeks vs. 52 sessions of HIIT in 9 weeks), our results are consistent with the concept that SC assemblies are a structural feature of the ETC rather than a phenomenon conferring enhanced bioenergetics[53]. In line with this, our data indicated that the greatest changes in maximal mitochondrial respiration ($[ETF + CI + II]_P$) occurred in parallel with the greatest changes in SC abundance (compare Fig. 5b with 5c, both upper panels). However, following normalisation by MPE, both of these adaptations were no longer significant (compare Fig. 5b, middle panels with Fig. 5c, lower panel). Taken together, our findings do not support the hypothesis that training-induced changes in the abundance or organisation of SCs contribute to improvements in mitochondrial respiration. Conversely, our data indicate that training-induced changes in both mitochondrial respiration and SC formation could largely be attributed to increases in mitochondrial content.

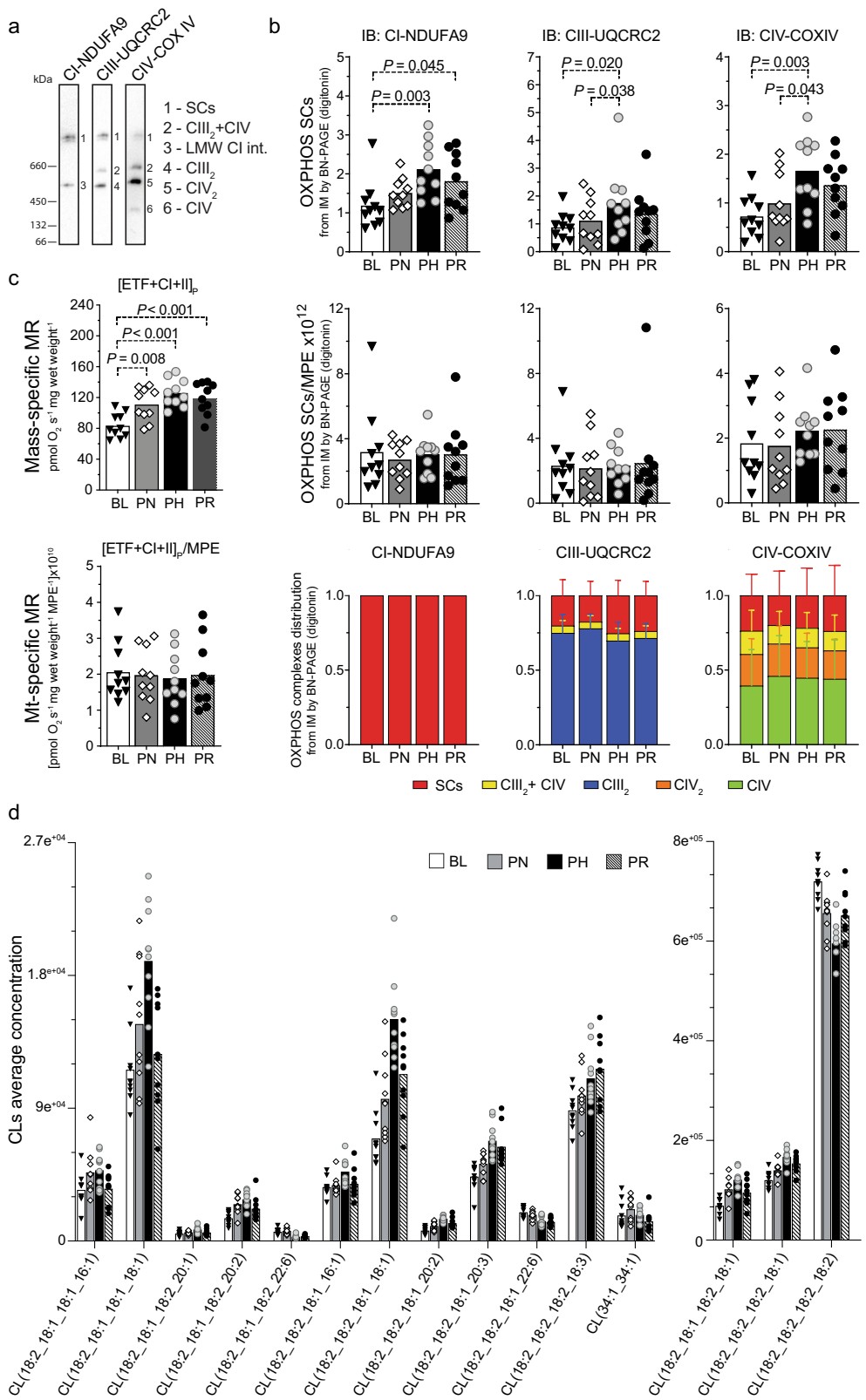

Given the importance of CLs for the assembly and stability of mitochondrial membrane protein complexes and ETC SCs[55,56], we again interrogated our normalised mitochondrial lipidome. Our analysis demonstrated an extensive remodelling of CL composition (Fig. 5d); notably, following NVT and HVT, we observed a decrease in the abundance of tetra-linoleoyl CL (18:2_18:2_18:2_18:2), the dominant form of CLs found in

skeletal muscle[57]. Accompanying this were concomitant increases in CLs containing oleic acid (18:1) and to a lesser extent palmitoleic acid (16:1) acyl chains. This may reflect a lack in relative availability of dietary linoleic acid due to increased mitochondrial biogenesis, since the nature of CL remodelling catalysed by TAZ, which was unaltered in our dataset, appears to be controlled predominantly by lipid availability[58]. This

**Fig. 5 Changes in mitochondrial content without reorganisation of SCs occur alongside improvements in mitochondrial respiration. a** Representative BN-PAGE blots of isolated mitochondria (IM) fractions from human vastus lateralis muscles (images from baseline [BL] IM fractions). Band 1 (SCs): mature supercomplexes (SCs) consisting of complex $I + III_n + IV_n$; band 2 ($CIII_2 + IV$): a SC consisting of CIII and CIV; band 3 (LMW CI int): low molecular weight intermediate of CI (band not present in all samples); band 4 ($CIII_2$): CIII dimer; band 5 ($CIV_2$): CIV dimer; band 6 (CIV): CIV monomer. **b** Top panels: protein content of SCs of the electron transport chain (ETC) by BN-PAGE in IM fractions from human vastus lateralis muscle biopsies (antibodies probed on separate membranes); middle panels: values from top panels normalised by mitochondrial protein enrichment (MPE); lower panels: distribution of ETC complexes into SCs from images obtained in b at each time point. Source data are provided as a Source Data file. All samples analysed were derived from the same experiment and blots were processed in parallel. **c** Top panel: maximal mass-specific mitochondrial respiration (MR) in permeabilised human vastus lateralis muscle fibres with convergent electron input through ETF + CI + CII ($[ETF + CI + II]_P$); lower panel: values of mitochondrial- (mt-) specific MR, obtained after normalising values from the top panel by MPE ($[ETF + CI + II]_P/MPE$). Results for the entire substrate-uncoupler-inhibitor-titration (SUIT) protocol are shown in Supplementary Fig. 1b. $P$ values are indicated on the figure to three decimal places; for $P$ values that were truncated, the corresponding accurate $P$ values were: PH vs. BL: $P = 5.2e^{-5}$; PR vs. BL: $P = 4.5e^{-4}$. Source data are provided as a Source Data file. **d** Training-induced changes in all individual cardiolipin (CL) species that were identified as differentially expressed (as determined in Fig. 4c, by linear model fit using *limma* with empirical Bayes method with an adjusted $P < 0.05$ [Benjamini–Hochberg]). Source data are provided as a Source Data file. PN post-NVT, PH post-HVT, PR post-RVT, IB immunoblotting. For panels **b**–**d** filled triangles, empty diamonds, and empty and filled circles represent individual values; the maxima of each bar represents the mean value; $n = 10$ for all analyses; datasets **b** and **c** were analysed by repeated measures one-way ANOVA followed by Tukey's post hoc testing, except for the lower right panel of **b**, which was analysed by Friedman test followed by Dunn's post hoc testing, as not normally distributed; $P < 0.05$.

---

compensation in CL content is in line with our observation that increased training volume does not lead to changes in ETC SC assembly (Fig. 5b middle and lower panels).

**Coenzyme Q biosynthesis and cristae formation are not prioritised during high-volume endurance training.** Despite enrichment analysis suggesting cluster-5 to be enriched in metabolism of amino acids and derivatives, upon further inspection we identified this cluster to predominantly contain proteins involved in the biosynthesis and function of coenzyme Q (CoQ or ubiquinone; Fig. 2f; Supplementary Fig. 2d and Supplementary Data 5) - an enzyme with an essential role as an electron transfer lipid in OXPHOS processes[59], which has been shown to be altered following exercise training in mouse skeletal muscle[54]. We identified changes in four constituents (COQ3, COQ5, COQ7, COQ9) of the CoQ synthome (Fig. 3 and Supplementary Data 6), a multi-subunit complex necessary for the biosynthesis of CoQ[60]. We report no changes post-NVT, consistent with previous research investigating the effects of short-duration (<3 weeks) exercise training on CoQ content in human skeletal muscle[61]. However, post-HVT we report a concerted reduction in three of these proteins, which also persisted post-RVT (Supplementary Data 7). This indicates that CoQ biosynthesis is not prioritised in response to high-volume endurance training, suggesting that enhancing the CoQ pool may not be necessary to support an increase in mitochondrial respiration in human skeletal muscle of young healthy individuals.

CoQ is an electron carrier shuttling electrons derived from the TCA cycle (via CI and CII), pyrimidine biosynthesis (via DHODH), glycolysis (via GPD2), and FAO (via EFTDH) to Complex III[45]. Moreover, other enzymes located near or on the inner mitochondrial membrane feeding electrons to CoQ, such as IVD[62], DLD[63] and L2HGDH[64], were also identified in our study (Supplementary Data 6). Similar to the decrease reported above in CI and CII subunits post-NVT, we report decreases in four of these enzymes during HVT, with three of them remaining downregulated both post-HVT and post-RVT; conversely, IVD was increased during NVT and DHODH was elevated post-HVT (Fig. 3 and Supplementary Data 7). The lack of a clear correlation between the content of the above proteins and that of CoQ may be ascribed to changing preferences for energy utilisation over the course of the training intervention, as also indicated by the differential prioritisation of the various metabolic pathways described above.

The mitochondrial contact site and cristae organising system (MICOS) is pivotal for the formation of cristae junctions, providing the extended membrane surface hosting OXPHOS complexes[65]. The MICOS associates with the outer membrane sorting and assembly machinery (SAM) to yield the mitochondrial intermembrane space bridging complex (MIB), thus linking the mitochondrial inner and outer membranes[65]. Eight proteins involved in cristae formation were differentially expressed in our study: APOO (MIC26), APOOL (MIC27), CHCHD3 (MIC19) and MIC60 from the MICOS complex; MTX1 and MTX2 from the SAM complex; TMEM11, a protein associating with multiple MICOS subunits for cristae biogenesis[66]; and OMA1, a mitochondrial protease regulating OPA1[67] - a key protein implicated in cristae remodelling[68] (Fig. 3 and Supplementary Data 6). Seven of these proteins, which almost entirely grouped within cluster-6 (Fig. 2f; Supplementary Fig. 2d and Supplementary Data 5), were significantly decreased post-HVT, with four being decreased post-RVT (Supplementary Data 7), suggesting a training-induced deprioritisation of these proteins. This is consistent with the minimal changes observed in cristae density following short-term (<3 months) exercise training in obese individuals[69]. Thus, our results support the notion that cristae remodelling is not a priority during short-term training and that changes in other mitochondrial proteins are more important to adapt to the higher metabolic demand of exercise. Despite MICOS-MIB subunits being known to interact with CLs[70], we report no correlation between the levels of MICOS-MIB subunits and changes in CLs content following different training volumes.

**Concluding remarks.** The present study utilised the power of multiple omics techniques, and an in silico normalisation strategy eliminating the bias introduced by changes in mitochondrial content, to unearth an intricate and previously undemonstrated network of differentially prioritised mitochondrial adaptations to exercise training in men. We identified 185 differentially expressed mitochondrial proteins, an ~10-fold greater number than previous studies[23,32,33]. This increased depth of analysis enabled us to identify the time-dependent and complex remodelling of the mitochondrial proteome following different phases of endurance exercise training. We demonstrate that training-induced changes in individual proteins, protein functional classes, and metabolic pathways follow markedly different patterns of adaptation, which are not stoichiometrically linked to the overall training-induced increase in mitochondrial content. Moreover, we show that the lack of stoichiometry and the differential

prioritisation extended to all three levels investigated (i.e., the transcriptome, proteome, and lipidome) and between levels, as demonstrated by the delayed adaptive response of lipids compared to proteins.

A striking and unexpected finding was the early (post-NVT) deprioritisation of OXPHOS subunit formation, which indicates that the overall training-induced increase in biogenesis and the proliferation of mitochondria is greater than the increase in specific components of the OXPHOS machinery. This deprioritisation of OXPHOS subunit biogenesis becomes even more striking when considering the early increase in mitochondrial respiration, as well as proteins involved in the TCA cycle and FAO processes - two of the main suppliers of reducing equivalents to the OXPHOS system. This increase could not be explained by changes in the abundance and/or organisation of SCs, as previously hypothesised[52], since we observed no change in these parameters. Our findings therefore add to the growing evidence that SC formation does not confer enhancements in mitochondrial bioenergetics[53]. The deprioritisation of OXPHOS subunit biogenesis is also unlikely to be related to the time required to assemble large multi-protein complexes[48], as the relative amount of OXPHOS subunits remained reduced throughout the intervention (lasting several weeks), whereas mitochondrial respiration, as well as TCA cycle and FAO process, remained upregulated throughout. Instead, the increase in mitochondrial respiration was likely supported by the inherent reserve capacity of mitochondria (i.e., their ability to respond to sudden increases in energy requirements)[71], which suggests that enhancing electron flow to OXPHOS is more important to increase ATP production than an increase in the components of the OXPHOS machinery. The relative decrease in proteins important for both CoQ biosynthesis and cristae formation further substantiates these findings.

Although the influence of training duration cannot be completely ruled out, the above findings were the result of changes in only one of the several exercise prescription programming variables (i.e., training volume). Future research should investigate the effects of manipulating other programming variables (e.g., exercise intensity, frequency and recovery between sessions), the type of exercise (e.g., running, swimming and resistance training), and different populations (e.g., differences in age, sex and health status) utilising a similar approach. These interventions are likely to induce adaptations that are specific and presumably different from the one presented here. Because of the well-documented therapeutic benefits of exercise training[72,73], the knowledge generated by our findings, which are readily available in supplemental tables, together with the results from future research, could be mined by exercise and health professionals to design focused and personalised training interventions.

## Methods

**Participants and ethics approval.** Ten healthy men volunteered to take part in this study (physiological and performance parameters are presented in Table 1). Potential participants were deemed suitable if aged 18–35 y, were moderately-trained (i.e., <4 h per week of unstructured aerobic activity for half a year prior to the study), not regularly engaged in cycling-based sports, and were non-smokers and medication free prior to and during the study. Participants underwent a medical screening to exclude conditions that may have precluded their participation (e.g., cardiovascular, musculoskeletal and/or metabolic problems), and were informed of the study requirements, risks, and benefits, before giving written informed consent. Approval for the study's procedures, which conformed to the standards set by the latest revision of the Declaration of Helsinki, was granted by the Victoria University Human Research Ethics Committee (HRE15-126).

**Study design.** The study consisted of three consecutive training phases: the normal- (NVT), high- (HVT) and reduced- (RVT) training volume phase (Fig. 1a). Each training phase was preceded (and followed) by performance testing, which included a 20-km cycling time trial (20k-TT), a graded exercise test (GXT) (participants were previously familiarised with both tests), and a resting muscle biopsy. Overall study duration was ~9 weeks.

**Testing procedures.** Participants were required to avoid any vigorous exercise for the 48 h preceding each performance test (72 h for the skeletal muscle biopsy), from alcohol and any exercise for 24 h before testing, and from food and caffeine consumption for the 2 h preceding each test. Similar tests were performed at the same time of the day throughout the study to avoid variations caused by changes in circadian rhythm.

*GXT.* A graded exercise test was performed on an electronically braked cycle ergometer (Lode Excalibur v2.0, Groningen, The Netherlands) to determine peak oxygen uptake ($\dot{V}O_{2Peak}$), peak power output ($\dot{W}_{Peak}$), the power attained at the lactate threshold ($\dot{W}_{LT}$) using the modified $D_{Max}$ method[74], and the training intensity for each training phase. The test consisted of consecutive 4-min stages at constant power output; the test starting intensity (range: 45–77 W) and the intensity increase of each stage (range: 17–28 W) were chosen so as to obtain at least 8 time points for the determination of the $\dot{W}_{LT}$[75] and were based on participants' fitness levels. An identical protocol was used at all four time points for each participant. Prior to the test, and in the last 30 s of each stage, venous blood samples were taken for measurement of blood lactate concentration ([La$^-$]). Participants were instructed to keep a cadence > 60 rpm and were only allowed access to cadence. The test was stopped when a participant reached volitional exhaustion or cadence dropped below 60 rpm for over 10 s. The $\dot{W}_{Peak}$ was determined as the power of the last completed stage plus an additional 25% of the stage increase wattage for every additional minute completed. At the end of each GXT, after a 5-min recovery, a verification exhaustive bout was performed at an intensity equivalent to $\dot{V}O_{2Peak}$ to confirm the highest measured $\dot{V}O_{2Peak}$[75].

*20k-TT.* Cycling time trials were performed on an electronically braked cycle ergometer (Velotron, RacerMate, Seattle, WA, USA) after a 6-min cycling warm-up (4 min at 66% of $\dot{W}_{LT}$ followed by 2 min at $\dot{W}_{LT}$), and 2 min of rest. Participants were only allowed access to cadence and completed distance.

*Gas Analysis during the GXT.* During the GXT, expired air was continuously analysed for $O_2$ and $CO_2$ concentrations via a gas analyser (Moxus 2010, AEI Technologies, Pittsburgh, PA, USA), which was calibrated immediately before each test. $\dot{V}O_2$ values were recorded every 15 s and the average of the two highest consecutive 15-s values was recorded as a participant's $\dot{V}O_{2Peak}$.

   *Venous blood sampling*: Venous blood samples (~1 mL) were collected during the GXT from a cannula inserted in the antecubital vain for the determination of venous blood [La$^-$] using a blood-lactate analyser (2300 STAT Plus; YSI, Yellow Spring, OH, USA).

*Muscle biopsies.* A biopsy needle with suction under local anaesthesia (1% xylocaine) was used to obtain vastus lateralis muscle biopsies at rest at the following four time points: BL, PN, PH and PR. After being cleaned of excess blood, connective and fat tissue muscle biopsies were divided as follows: ~10 mg was immediately immersed in ~2 mL of ice-cold BIOPS for measurements of mitochondrial respiration, whereas the remainder was promptly frozen in liquid nitrogen and stored at −80 °C for follow-up analyses.

**Training intervention.** All training sessions were performed on an electronically braked cycle ergometer (Velotron, RacerMate, USA), following an 8-min warm up (see 20k-TT) and consisted of HIIT (2:1 work-to-rest ratio). Training intensity was set relative to $\dot{W}_{LT}$ (rather than $\dot{W}_{Peak}$) so as to induce similar metabolic and cardiac stresses amongst participants of differing fitness levels[76]. Exercise intensity was maintained between $\dot{W}_{LT}$ and $\dot{W}_{Peak}$ throughout the entire study so that training volume was the only manipulated variable between the three phases.

*NVT phase.* This consisted of 6 HIIT sessions within 2 weeks of 5–7 4-min cycling intervals interspersed with a 2-min recovery at 60 W. Exercise intensities were defined as [$\dot{W}_{LT} + x(\dot{W}_{Peak}-\dot{W}_{LT})$], with $x$ increasing from 0.5 to 0.7 throughout the phase.

*HVT phase.* Participants performed HIIT twice a day for 20 consecutive days; training sessions consisted of either 7–10 4-min intervals interspersed with a 2-min recovery at 60 W at intensities ranging from [$\dot{W}_{LT} + 0.5(\dot{W}_{Peak}-\dot{W}_{LT})$] to [$\dot{W}_{LT} + 0.8(\dot{W}_{Peak}-\dot{W}_{LT})$], or 15–20 2-min intervals at intensities ranging from [$\dot{W}_{LT} + 0.5(\dot{W}_{Peak}-\dot{W}_{LT})$] to [$\dot{W}_{LT} + 0.95(\dot{W}_{Peak}-\dot{W}_{LT})$], interspersed with a 1-min recovery at 60 W. Single session duration increased from ~45 min to 60 min.

*RVT phase.* The RVT phase consisted of 6 HIIT sessions in 6 days; participants performed 10, 9, 8, 7, 6, and 4, 4-min intervals interspersed with a 2-min recovery at 60 W, at an intensity of [$\dot{W}_{LT} + x(\dot{W}_{Peak}- \dot{W}_{LT})$], with $x$ increasing from 0.5 to 0.7 throughout the phase.

**Physical activity and nutritional control.** Physical activity and dietary patterns were maintained throughout the study and were monitored with the use of food and physical activity recall diaries. The last three meals prior to each performance test undertaken during baseline testing were recorded by each participant and were

replicated thereafter before the same type of test. To control for dietary effects on muscle metabolism, participants were provided with a standardised dinner (55 kJ kg$^{-1}$ body mass (BM), providing 2.1 g carbohydrate (CHO) kg$^{-1}$ BM, 0.3 g fat kg$^{-1}$ BM, and 0.6 g protein kg$^{-1}$ BM) and breakfast (41 kJ kg$^{-1}$ BM, providing 1.8 g CHO kg$^{-1}$ BM, 0.2 g fat kg$^{-1}$ BM, and 0.3 g protein kg$^{-1}$ BM), to be consumed 15 and 3 h prior to the muscle biopsy, respectively.

## Muscle analyses

*Enzymatic activity.* Enzyme activities were determined spectrophotometrically in post-600g supernatants of skeletal muscle homogenates according to the method described for respiratory chain complexes I–IV and citrate synthase[77]. Briefly, ~20 mg of skeletal muscle sample was homogenised in sucrose/mannitol containing buffer using a glass/glass homogeniser and spun for 10 min at 600 g and 4 °C. The supernatant was then subjected to two freeze/thaw cycles and stored at −80 °C until measurement of respiratory chain enzymes. CI was assayed as rotenone-sensitive NADH:CoQ1 oxidoreductase by monitoring the decrease in absorbance due to NADH oxidation at 340 nm. For CII, activity was measured as succinate:CoQ1 oxidoreductase by measuring CoQ1 reduction at 280 nm. CIII was assayed as decylbenzylquinol; cytochrome *c* oxidoreductase by following the increase in absorbance resulting from cytochrome *c* reduction at 550 nm. CIV was measured as cytochrome *c* oxidase by following the decrease in absorbance resulting from cytochrome *c* oxidation at 550 nm. To assay the CS catalysed production of coenzyme A (CoA.SH) from oxaloacetate, the generation of free sulfhydryl groups was monitored using the thiol reagent 5,5'-dithio-bis-(2-nitro-benzoic acid) (DTNB), which reacts spontaneously with the sulfhydryl groups to produce 5-thio-2-nitrobenzoate anions. CS specific activity was measured by following the increase in absorbance resulting from the formation of 5-thio-2 nitrobenzoate anions at 412 nm. Following the enzyme measurements, the amount of protein in each sample was determined using a bicinchoninic acid assay and activity calculated as initial rates (complexes I, II and citrate synthase) or as first-order rate constants (complexes III and IV).

*Preparation of whole-muscle lysates for SDS-PAGE assessment of ETC subunits.* Frozen skeletal muscle samples (~10 mg) were homogenised in a TissueLyser for 2 × 2 min at maximum speed in ice-cold lysis buffer (1:20 w/v) containing 50 mM Tris-HCl, 150 mM NaCl, 1 mM EDTA, 1% NP-40 and a phosphatase/protease inhibitor (5872, Cell Signaling Technology, Danvers, MA, USA). Homogenates were rotated end-over-end at 4 °C for 1 h and protein concentration was determined in triplicate using a commercial colorimetric assay (Bio-Rad Protein Assay kit-II, Australia).

*Mitochondrial isolation for SDS- and BN-PAGE assessment of ETC subunits, complexes and SCs.* Frozen skeletal muscle samples (~30 mg) were homogenised with 2 × 20 strokes in a Potter-Elvehjem tissue grinder attached to a rotating drill (~1000 rpm) in 5 mL solution A (1 mM EDTA, 220 mM mannitol, 20 mM HEPES-KOH [pH = 7.6], 70 mM sucrose, 2 mg/mL BSA, 0.5 mM PMSF) and spun at 800 × *g* for 5 min at 4 °C. The supernatant was collected, whereas the pellet was re-homogenised as above in 5 mL of solution A to maximise extraction. The two supernatants were mixed and further spun at 800 g for 5 min at 4 °C. The ensuing supernatant was then spun at 10,000 × *g* for 20 min at 4 °C, and the pellet was resuspended in 200 μL of sucrose buffer (0.5 M sucrose, 10 mM HEPES-KOH [pH = 7.6], 0.5 mM PMSF). Protein concentration was determined by the bicinchoninic acid method according to the manufacturer's instructions (BCA Protein Assay Kit, Pierce-Thermo Fisher Scientific, Melbourne, Australia). This measurement was used to generate Fig. 1c.

*SDS-PAGE.* Both whole-muscle lysates (7.5 μg) and mitochondrial isolates (5 μg) were separated by electrophoresis using 12 or 15% SDS-PAGE gels, as previously described[78], and blotted with a total OXPHOS (ab110411, Abcam, Cambridge, MA, USA) or with a single CI (ab110242, Abcam, Cambridge, MA, USA) antibody where separation with CIV was not optimal using the total OXPHOS antibody. All antibodies were used at the final concentration suggested by the manufacturer (7.2 μg/mL [ab110411] and 0.5 μg/mL [ab110242]).

*BN-PAGE.* Mitochondrial isolates (6–15 μg) were separated by electrophoresis using 3–12% NativePAGE gels (Life Technologies Australia, Mulgrave, Australia) as previously described[79]. A 4 g/g digitonin/protein ratio was used for assessment of SCs. The following primary antibodies were used: NADH:ubiquinone oxidor-eductase subunit A9 (NDUFA9; ab14713), ubiquinol-cytochrome *c* reductase core protein 2 (UQCRC2; ab14745) and cytochrome *c* oxidase subunit IV (COX IV; ab14744) (all Abcam, Cambridge, MA, USA). All antibodies were used at the final concentration suggested by the manufacturer (1 μg/mL for all three antibodies used).

For both SDS- and BN-PAGE, protein bands were visualised using a Bio-Rad ChemiDoc imaging system and bands were quantified using Bio-Rad Image Lab 5.0 software (Bio-Rad laboratories, Gladesville, NSW, Australia). An internal standard (made of a mixture of all samples) was loaded in each SDS- and BN-PAGE gel, and each lane was normalised to this value, to reduce gel-to-gel variability.

*Fibre preparation for high-resolution respirometry.* Fresh muscle fibres were mechanically separated in ice-cold BIOPS (in mM: 2.77 CaK$_2$EGTA, 7.23 K$_2$EGTA, 5.77 Na$_2$ATP, 6.56 MgCl$_2$, 20 taurine, 50 MES, 15 Na$_2$phosphocreatine, 20 imidazole and 0.5 dithiothreitol adjusted to pH 7.1[13], followed by permeabilisation by gentle agitation for 30 min at 4 °C in BIOPS containing 50 μg/mL of saponin, and three 5-min washes in MiR05 (in mM, unless specified: 0.5 EGTA, 3 MgCl$_2$, 60 K-lactobionate, 20 taurine, 10 KH$_2$PO$_4$, 20 HEPES, 110 sucrose and 1 g/L BSA essentially fatty acid-free, pH 7.1)[13]. Mitochondrial respiration was measured in duplicate (from 2 to 3 mg wet weight of muscle fibres) in MiR05 at 37 °C using the high-resolution Oxygraph-2k (Oroboros, Innsbruck, Austria). To avoid potential oxygen diffusion limitation, oxygen concentration was maintained between 270 and 480 nmol mL$^{-1}$ by re-oxygenation via direct syringe injection of O$_2$.

*Mitochondrial respiration protocol.* The substrate-uncoupler-inhibitor titration (SUIT) protocol[13] used was as follows: 0.2 mM octanoylcarnitine and 2 mM malate ([ETF]$_L$: leak respiration state [L] in the absence of adenylates and limitation of flux by electron input through electron transfer flavoprotein [ETF]); 3 mM MgCl$_2$ and 5 mM ADP ([ETF]$_P$: maximal OXPHOS capacity [P] with saturating levels of ADP and limitation of flux by electron input through ETF); 5 mM pyruvate ([ETF + CI]$_P$: P with saturating levels of ADP and limitation of flux by convergent electron input through ETF + CI); 10 mM succinate ([ETF + CI + II]$_P$: P with saturating levels of ADP and limitation of flux by convergent electron input through ETF + CI + CII); 10 μM cytochrome *c* (outer mitochondrial membrane integrity test); 0.75-1.5 μM carbonyl cyanide 4-(trifluoromethoxy) phenylhy-drazone (FCCP) via stepwise titration ([ETF + C + II]$_E$, maximal electron transport chain capacity [E] with saturating levels of ADP and limitation of flux by convergent electron input through ETF + CI + CII); 0.5 μM rotenone ([CII]$_E$: E with saturating levels of ADP and limitation of flux by electron input through CII); and 5 μM antimycin A (residual non-mitochondrial oxygen consumption [ROX]). Data are presented as mass-specific mitochondrial respiration [pmol O$_2$ s$^{-1}$ mg$^{-1}$ wet weight] and as mitochondrial-specific respiration [pmol O$_2$ s$^{-1}$ mg$^{-1}$ wet weight/CS activity].

*RNA-seq analysis.* Approximately 10–15 mg of frozen muscle was used to isolate RNA using the RNeasy Mini Kit (Qiagen, Canada) according to the manufacturer's instructions. Samples were homogenised using the TissueLyser II (Qiagen, Canada). RNA concentration and purity were determined spectrophotometrically (NanoDrop 2000, Thermo Fisher Scientific, Wilmington, DE, USA) at 260 and 280 nm. RNA integrity was assessed using an Agilent Bioanalyser according to manufacturer's instructions. The RNA was stored at −80 °C.

*Sequencing and assembly of RNA-seq.* This analysis was conducted on n = 5 participants (20 samples in total); samples were sequenced (100 base pair, single reads) on the Illumina NovaSeq 6000 platform at the Australian Genome Research Facility (AGRF). Transcriptome assembly was completed at AGRF with reads screened for presence of any Illumina adaptor/over-represented sequences and cross-species contamination. Per base sequence quality for all samples was >96% bases above Q30. Cleaned sequence reads were aligned against the *Homo sapiens* genome (Build version HG38). The STAR aligner (v2.5.3a) was used to map reads to the genomic sequences. Counts of reads mapping to each known gene were summarised to provide the matrix used for further analysis.

*Bioinformatic analysis of RNA-seq data.* For downstream RNA-seq analysis the R package *limma*[80] was used to conduct the differential expression analysis from count data. Count data was normalised using calcNormFactors in the *edgeR* package in R. Linear modelling was performed in *limma* to identify transcripts that were differentially expressed across the three pairwise comparisons (i.e., PN vs. BL, PH vs. PN, and PR vs. PH). The resulting differential expression values were filtered for an adjusted *P* value < 0.05 using the Benjamini–Hochberg method. Differentially expressed transcripts were visualised in the heatmap in Fig. 1e using hierarchical clustering with the "average" method. Categorical columns for the generation of Fig. 1f were determined by identifying the transcripts to known descriptions in the literature; MICOS[81], TCA cycle[82], SLC25A[44] and Assembly Factors[83]. An enrichment analysis using Enrichr [https://maayanlab.cloud/Enrichr/[84,85]] was then performed on the same list of differentially altered transcripts with the results shown in Fig. 1g and Supplementary Data 2.

*Mitochondrial isolation for proteomics and lipidomics assessment.* Frozen skeletal muscle samples (~30 mg) were homogenised with 2 × 20 strokes in a Potter-Elvehjem tissue grinder attached to a rotating drill (~1000 rpm) in 3 mL solution B (1 mM EDTA, 220 mM mannitol, 20 mM HEPES-KOH [pH = 7.6], 70 mM sucrose, 0.5 mM PMSF) and spun at 1000 × *g* for 5 min at 4 °C. The supernatant was further spun at 12,000 × *g* for 10 min at 4 °C, and the ensuing pellet was resuspended in 200 μL of sucrose buffer (0.5 M sucrose, 10 mM HEPES-KOH [pH = 7.6]). Protein concentration was determined by the bicinchoninic acid method according to the manufacturer's instructions (BCA Protein Assay Kit, Pierce-Thermo Fisher Scientific, Melbourne, Australia). Two IM fractions per sample (50 μg each) were spun at 12,000 × *g* for 10 min at 4 °C; after removal of the supernatant, the pellets were stored frozen at −80 °C for subsequent proteomics and lipidomics analysis.

*Proteomics*. IM fractions were prepared for proteomics analysis as previously described[86] with minor modifications. Briefly, 50 μg of frozen mitochondrial isolates were solubilised in 20 μL of 8 M urea, 40 mM chloroacetamide, 10 mM tris(2-carboxyethyl)phosphine, 100 mM Tris, pH 8.1; this was followed by 15 min of sonication in a water bath sonicator and 30 min of shaking (1500 rpm, at 37 °C). The urea concentration was reduced to 2 M with $H_2O$ prior to protein digestion with trypsin (Promega, Alexandria, NSW, Australia) at a 1:60 trypsin:protein ratio and subsequent overnight digestion at 37 °C. The next day, samples were acidified with trifluoroacetic acid (1% [v/v] final concentration) and centrifuged for 5 min at $20,100 \times g$ at RT. The supernatants were desalted on pre-activated (100% acetonitrile [ACN]) and pre-equilibrated (0.1% TFA, 2% ACN) Empore™ SPE Disks (matrix active group polystyrene-divinylbenzene [SDB-XC] Merck, Bayswater, VIC, Australia) stage tips[87] made in-house, before being washed (0.1% TFA, 2% ACN) and eluted in 0.1% TFA, 80% ACN. Samples were concentrated under vacuum and reconstituted in 0.1% TFA, 2% ACN. After 15 min sonication and subsequent vortexing samples were centrifuged at $20,100 \times g$ at RT before estimation of peptide concentration (Direct Detect, Merck). Approximately 600–800 ng of peptides were analysed on a Thermo Q Exactive™ Plus mass spectrometer coupled to an Ultimate 3000 HPLC (both Thermo Fisher Scientific, Mulgrave, VIC, Australia). Peptides were first loaded onto a trap column (Dionex-C18, 100 Å, 75 μm x 2 cm; Thermo Fisher Scientific) at an isocratic flow of 5 μL min$^{-1}$ of 2% (v/v) ACN containing 0.1 % (v/v) formic acid (FA) for 5 min before switching the trap-column in line with the analytical column (Dionex-C18, 100 Å, 75 μm x 50 cm; Thermo Fisher Scientific). The separation of peptides was performed at 300 nL min$^{-1}$ using a nonlinear ACN gradient of buffer A (2% ACN, 0.1% FA) and buffer B (80% ACN, 0.1% FA) over 125 min including void and equilibration. Data were collected in positive mode using Data Dependent Acquisition using *m/z* 375–1400 as MS scan range, HCD for MS/MS of the 15 most intense ions with $z \geq 2$. Other instrument parameters were: MS1 scan at 70,000 resolution (at 200 *m/z*), MS maximum injection time 50 ms, AGC target 3e6, stepped normalised collision energy of 27, 30, 32, isolation window of 1.6 *m/z*, MS/MS resolution 17,500, MS/MS AGC target of 5e4, MS/MS maximum injection time 50 ms, minimum intensity was set at 2e3 and dynamic exclusion was set to 30 s.

Raw files were analysed using the MaxQuant platform[88] version 1.6.1.0, searching against the Uniprot human database containing reviewed, canonical variants in FASTA format (June 2018) and a database containing common contaminants by the Andromeda search engine[89]. Default search parameters for a label-free quantification (LFQ) experiment were used with modifications. In brief, "Label-free quantification" was set to "LFQ" using a minimum ratio count of 2. Cysteine carbamidomethylation was used as a fixed modification, and N-terminal acetylation and methionine oxidation were used as variable modifications. False discovery rates of 1% for proteins and peptides were applied by searching a reverse database, and 'match from and to', 'match between runs' options were enabled with a match time window of 0.7 min. Unique and razor peptides with a minimum ratio count of 2 were used for quantification.

*Bioinformatic analysis of proteomics data.* The R package *limma*[80] was used to conduct the differential expression analysis of MaxQuant LFQ intensities (extracted from proteinGroups.txt) after first performing normalisation using variance stabilising normalisation (VSN) as found in the *limma* package. Identifications labelled by MaxQuant as 'only identified by site', 'reverse' and 'potential contaminant' were removed. Proteins having <70% valid values were removed and remaining missing data was imputed using QRILC method from the *imputeLCMD* package in R[90]. Differential expression analysis was performed between BL and then each subsequent time point to show comparative changes between each of the training volumes, though comparisons across all three training phases were accounted for within the analysis. Linear modelling was determined using eBayes in the *limma* package. Proteins that were differentially expressed across all three training phases were identified by filtering for an adjusted *P* value < 0.01 using the Benjamini–Hochberg method. For the mitochondrial normalisation, all "Known Mitochondrial" proteins identified using the Integrated Mitochondrial Protein Index (IMPI)[29] were subset from the rest of the dataset, and then followed the normalisation and statistical validation as described above. Heatmaps were produced using hierarchical clustering using the "complete" method. Gene ontology of the clusters was determined by taking the proteins identified in the cluster and performing an enrichment analysis using the ClueGO (v2.5.6) application in Cytoscape (v3.7.1) using default settings except for a GO tree interval of 3 to 5, with only the Biological Processes Ontology switched on (Fig. 2f and Supplementary Fig. 2a). Full tables of identified proteins and post-normalisation differentially expressed proteins in their respective clusters and their annotations can be found in Supplementary Data 3–7.

The heatmaps presented in Fig. 3 and discussed in the Results section were generated according to Reactome pathways and/or literature searches (Supplementary Data 6). Proteins involved in two or more pathways were either presented in both pathways, or were assigned to the pathway involving the protein's primary function and/or most closely matching their training-induced changes. Specifically, "mitochondrial translation" was based on Reactome pathway R-HSA:5368287 (Mitochondrial translation). "TCA cycle" was based on Reactome pathway "R-HSA:71403 (Citric acid cycle [TCA cycle]); NNT was removed because its main physiological function is the generation of NADPH[91]. "FAO" was based on Reactome pathway R-HSA:556833 (Metabolism of Lipids); ETFA[92], ETFB[92],

HSD17B10[93], and PLIN5[42] were added based on their involvement with FAO and lipid metabolism; GPD2[45,94], GPX1 [R-HSA-3299685 and[46]], and GPX4[46] where not presented within this pathway as their primary function suggested their inclusion in different pathways of Fig. 3; SAR1B was removed because its main function is the regulation of vesicle budding[95]. "OXPHOS" was based on Reactome pathway R-HSA:1632000 (Respiratory electron transport, ATP synthesis by chemiosmotic coupling, and heat production by uncoupling proteins); ETFA[92], ETFB[92], ETFDH[45], TRAP1[49], and CYCS (R-HSA-3299685) were not presented within this pathway as their primary function and/or training-induced changes suggested their inclusion in different pathways of Fig. 3 was more appropriate; based on the main function of its constituent proteins the "OXPHOS" pathway was then subdivided in 2 subgroups: "OXPHOS - subunits", to which MP68[96] was added, and "OXPHOS - Assembly factors" to which ATPAF1[14], BCS1L[14], and COA3[14] were added. "Response to ROS" was based on Reactome pathway: R-HSA-3299685 (Detoxification of Reactive Oxygen Species); GPX4 was added based on its ability to reduce $H_2O_2$[46]. "Protein Folding" was based on literature searches: TRAP1, HSPA9, HSPD1, HSPE1, and TIMM44 were all added based on their involvement in protein folding and quality control mechanisms[49]. "CoQ biosynthesis" was based on Reactome pathway R-HSA-2142789 (Ubiquinol biosynthesis). "CoQ e- donors" was based on literature searches: DHODH[45], ETFDH[45], GPD2[45], IVD[62], DLD[63] and L2HGDH[64], were all added based on their function to feed electrons into the respiratory chain via CoQ. "Cristae formation" was based on Reactome pathway R-HSA-8949613 (Cristae formation); OMA1[67,68] was added based on its involvement in cristae formation, whereas ATP5A1, ATP5J, ATP5B, ATP5H, and ATP5O (all based on R-HSA:163200), as well as HSPA9[49] were not presented within this pathway as their primary function suggested their inclusion in different pathways of Fig. 3.

*Lipid extraction for lipidomics.* Mitochondrial isolates were extracted using a modified single-phase chloroform/methanol extraction as described previously[97]. In brief, 20 volumes of chloroform:methanol (2:1) were added to the sample along with a series of internal standards. Samples were vortexed and centrifuged on a rotary mixer for 10 min. Following sonication on a sonicator bath for 30 min, samples were rested for 20 min prior to centrifugation at $13,000 \times g$ for 10 min. Supernatants were transferred into a 96-well plated, dried down and reconstituted in 50 μL $H_2O$ saturated butanol and sonicated for 10 min. After the addition of 50 μL of methanol with 10 mM ammonium formate, samples were centrifuged at 4000 rpm on a plate centrifuge and transferred into glass vials with inserts for mass spectrometry analysis.

*Targeted lipidomics analysis.* LC-MS/MS was performed according to previously published methods, with slight modification for tissue samples[98]. Sample extracts were analysed using either (i) an AB Sciex Qtrap 4000 mass spectrometer coupled to an Agilent 1200 HPLC system for CL assessment, as described preciously[99] or (ii) an Agilent 6490 QQQ mass spectrometer coupled with an Agilent 1290 series HPLC system for assessment of all other lipid species[98]. Lipids run on the Agilent 6490 were measured using scheduled multiple reaction monitoring with the following conditions: isolation widths for Q1 and Q3 were set to "unit" resolution (0.7 amu), gas temperature 150 °C, nebuliser 20 psi, sheath gas temperature 200 °C, gas flow rate 17 L/min, capillary voltage 3500 V and sheath gas flow 10 L/min. The list of multiple reaction monitoring (MRM) used and the chromatographic conditions were described previously[98]. Chromatographic peaks were integrated using Mass Hunter (B.09.00, Agilent) and assigned to lipid species using MRMs and retention time, with quantification derived from the ratio of each peak with its corresponding internal standard.

*Bioinformatic analysis of lipidomics data.* For lipidomics we identified that the increase in CLs in the un-normalised data was due to a training-induced increase in MPE in our mitochondrial isolates. To eliminate this bias, all lipid species were normalised by total CL amount (Supplementary Data 8). Lipid species were log transformed before undergoing differential expression analysis in *limma*[80], with linear models using the eBayes function. The resulting differential expression values were filtered for an adjusted *P* value < 0.05 using Benjamini–Hochberg method (Supplementary Data 9). Heatmaps were visualised using hierarchical clustering using the "complete" method. Profile plots were determined by taking the mean of the *z*-score for all participants' samples for each lipid species at each time point (Fig. 4d). Forest plots are depicted using LogFC results from *limma* analysis (Supplementary Fig. 3). Higher classes were determined through descriptions of the literature as assigned by the Lipid Metabolites and Pathways Strategy (Lipid MAPS)[100].

*Statistical analysis.* All values are reported as means ± SD, unless otherwise specified. For non-omics analyses: outliers were first removed using the ROUT method set at $Q = 1\%$[101]. Normally distributed datasets (Shapiro–Wilk test $P > 0.05$) were analysed by a repeated measures one-way ANOVA followed by Tukey's correction post hoc testing. Non-normally distributed datasets (Shapiro–Wilk test $P < 0.05$) were transformed using, in order, log(Y), 1/Y, √Y, until normality was met, before being analysed as above; datasets that remained non-normal following three independent transformation attempts were analysed using the non-parametric Friedman test on the raw data followed by Dunn's correction post hoc testing. The

level of statistical significance was set a priori at $P < 0.05$. GraphPad Prism (v. 8.4.2) was used for all statistical non-omics analyses. Differential expression for all omics was completed in R (v.3.6.3), using the packages, *limma* and edgeR to determine significantly enriched results between the groups. Significance was determined by an adjusted $p$ value < 0.05 for the transcriptomics and lipidomics and 0.01 for the proteomics. Statistical details can be found in the associated figure legends. All figures were prepared using Adobe Illustrator (CC2018.22.1).

**Reporting summary**. Further information on research design is available in the Nature Research Reporting Summary linked to this article.

## Data availability

Source data to interpret, verify and extend this research is provided with this paper. The mass spectrometry proteomics data has been deposited in the ProteomeXchange Consortium via the PRIDE partner repository under accession code PXD026219. The transcriptomic data has been deposited in the NCBI and can be found under the BioProject PRJNA732106. The lipidomics data has been deposited in the NIH Common Fund's National Metabolomics Data Repository (NMDR) website, the Metabolomics Workbench, under Project ID ST001907 [https://doi.org/10.21228/M8G69Q]. Source data are provided with this paper.

## Code availability

The R scripts used for all omics analyses described above are deposited on GitHub and available through https://doi.org/10.5281/zenodo.5576974. There are no restrictions placed on accessibility of this code.

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

## Acknowledgements

We thank the participants for their time and commitment to this study. The authors would like to acknowledge all members of the Stroud and Bishop labs for input into interpretation and presentation of data. We thank the Bio21 Mass Spectrometry and Proteomics Facility (MMSPF) for provision of instrumentation, training and technical support. We acknowledge the use of the services and facilities of the Australian Genome

Research Facility (AGRF). This study was funded by grants from the ANZ-MASON Foundation (to D.J.B), the Australian Research Council (Discovery Project DP140104165 to D.J.B), the Australian National Health and Medical Research Council (NHMRC Project Grant 1140906 to D.A.S.; NHMRC Fellowships 1140851 to D.A.S. and 1155244 to D.R.T.), JDRF Australia (JDRF Career Development to M.T.C.), and the Australian Research Council Special Research Initiative in Type 1 Juvenile Diabetes (to M.T.C.). We acknowledge the support of the Mito Foundation for the provision of instrumentation and the Victorian Government's Operational Infrastructure Support Program.

## Author contributions

C.G., D.A.S. and D.J.B. conceptualised the study. C.G., D.A.S. and D.J.B. devised the study methodology. C.G., N.A.J. and H.A.J. performed study validation. C.G., N.A.J. and H.A.J. delivered the training and performed sample collection. C.G., J.B., J.K., B.R., A.L., T.L.S., and A.E.F. performed biochemical analyses. C.G., B.R., and D.A.S., performed proteomic analysis. K.H. and N.A.M. performed lipidomic analysis. RNA-seq analysis was performed at the Australian Genome Research Facility (AGRF). C.G., N.J.C. and D.A.S. performed statistical and bioinformatic analysis. C.G., N.J.C. and D.A.S. delivered the visualisation. C.G., D.A.S. and D.J.B. wrote the manuscript. M.T.C., P.J.M., D.R.T., D.A.S. and D.J.B. provided supervision. D.R.T., M.T.C., D.A.S. and D.J.B. funded the research. C.G., D.A.S. and D.J.B. have primary responsibility for final content. Data collection took place at Victoria University. Muscle analysis took place at Victoria University, Monash University, Murdoch Children's Research Institute, the Baker Heart & Diabetes Institute, AGRF, and the Bio21 Molecular Science & Biotechnology Institute (The University of Melbourne). All persons designated as authors qualify for authorship, and all those qualifying for authorship are listed. All authors have read and approved the final manuscript.

## Competing interests

The authors declare no competing interests.
