## [Peer Review File · Nature Communications]

High-intensity training causes non-stoichiometric changes in the mitochondrial proteome of human skeletal muscle without reorganisation of respiratory chain contentREVIEWER COMMENTS

Reviewer #1 (Remarks to the Author):

The authors performed multi-omic analyses in a well-controlled study of human exercise training. The authors should be commended for this. They undertake a rigorous analysis of the mitochondrial proteome, including a simple but effective means of normalizing the data for exercise-induced mitochondrial biogenesis. These are really impressive data and I recommend that this manuscript should be published with only minor modifications:

Supplemental Fig 1: Please show representative images of immunoblots

Fig 1F: Colour could be incorporated to better highlight the profiles of the selected proteins, as in 2C and 4D.

Fig 1G: Are the transcriptomics enrichment analyses performed as an “average” of the three post-training timepoints? Please describe this method in further detail (in the methods section).

Lines 130-133: When normalized to CS activity many of the increases in respiratory states and complex activity, which are apparent as absolute values, are abolished in the PH condition. Please highlight and discuss this in text in relation to effects of training duration, volume and intensity on mitochondrial content and efficiency. At least refer to similar observations you made in Granata et al (2016) (<https://pubmed.ncbi.nlm.nih.gov/27402675/>).

Line 166: Whilst I appreciate the validity and utility of the transcriptomic data, transcriptomic data cannot provide an insight into protein stoichiometry. Please amend the sentence on lines 166-167.

Line 175: Only the top 5 enriched GOBP terms are displayed in Fig 1g, therefore it is impossible for readers to verify that mitochondrial terms were not enriched. Please display all of the enrichment terms, for example in a bubble plot, or, if there are too many to display, please refer to a supplemental table

Figs 2E, 3 & 4C: Whilst I can interpret the heat maps shown in these figures, in which the fold change of each training block is compared to the previous, I would find it easier to follow the time-course of the adaptations if each training block was compared to baseline.

Pages 27, 28 and 29: Reference should be made to the (very) newly published study from Gonzalez-Franquesa et al (2021) (<https://pubmed.ncbi.nlm.nih.gov/34038727/>), in which SC formation and COQ protein complex formation was studied in muscle in the context of exercise training, albeit in mice.

Results & Discussion: I find the order of the results is illogical – specifically it seems strange to jump from proteomics (fig 3) to lipidomics (fig 4) and back again. In my opinion, the results and discussion pertaining to COQ biosynthesis and cristae formation should also come earlier.

Lines 626-627: It should be considered that some of the changes seen may be due to timing (i.e. duration of the training programme) rather than training volume – after all two weeks is a relatively short duration and some exercise-induced adaptations may take longer. For example, translatory proteins may be one of the early adaptations to exercise training/an increase in volume, whilst the expression of OXPHOS complex proteins may follow.

Reviewer #2 (Remarks to the Author):

The manuscript by Granata et al. uses a combination of biochemical, transcriptomic, proteomic and lipidomic approaches to evaluate changes on skeletal muscle mitochondria after a period of exercise training in humans. To distinguish the effects of overall increased mitochondrial content after exercise training from more specific mitochondrial adaptations, the authors apply an *in silico* normalization to both proteomics and lipidomics data. This allowed the identification of protein/lipid clusters that the authors propose to be differentially prioritized during exercise training.

After observing that changes in mitochondrial constituents are not stoichiometrically linked to the overall mitochondrial content, the authors show that TCA cycle and FAO are prioritized over the expression of OXPHOS subunits. This, together with the data on mitochondrial supercomplexes, led them to conclude that during exercise training enhancing electron flow to OXPHOS is more important for ATP generation than the overall OXPHOS content.

The study is very well designed and represents a clear sophistication of the group's research efforts, which are very well known in the field of exercise physiology. The mitochondrial phenotyping is impressive, especially considering the logistics involved in collecting the samples and the limited amount of material to work with.

Conceptually, the study addresses a long-standing problem of how to decouple mitochondrial content from respiratory function. Although the data herein presented constitute an important step towards solving this issue, the conclusions put forward in the manuscript are quite strong (sometimes provocative) and not entirely supported by the data, remaining mostly associative and descriptive. Indeed, that might be the main shortcoming of the work, as it sets the frame for the identification of novel players in mitochondrial function, but then it concludes without really advancing our understanding of how skeletal muscle mitochondria adapt to exercise training from a mechanistic perspective. This is emphasized by the fact that previous research, including work from the same group, has already shown some disconnect between mitochondrial content and function (Granata et al 2016), highlighting the importance of TCA cycle and FAO for early adaptations to exercise training.

Here are some points that could improve the general impact and message of the study.

I don't think it's fair to call this a multi-omics analysis (rather a "multiple-omics") as the different omics data are not analyzed in a truly integrated way. This would require some additional (and not trivial) bioinformatics analysis, where all the different layers of data are co-analyzed to identify connections and correlations between them. This would definitely improve the study's impact, but this type of analyses are a project in itself.

Contrary to what the authors had observed before (Granata 2016), a reduction of markers for mitochondrial content post-RVT was not seen. This could be explained by a shorter RVT period and/or increased volume compared with the previous study (6 sessions over 1 week vs 5 sessions over 2 weeks). The authors should comment on how this could affect the results herein presented, especially considering the mitochondrial proteome rearrangement. Would it be expected that OXPHOS proteins decrease (prioritized removal) faster than TCA- and FAO-related proteins?

Since in essence this is a resource paper, the transcriptome analysis could have been better explored, e.g., comparing the changes observed in the mitochondrial proteome with corresponding transcripts. Is there a clear disconnect between transcripts and proteins? A heatmap showing the transcriptional changes of the same genes presented in Fig 2e could be helpful (highlighting those that achieved significance). Taking advantage of the MitoPathways annotations recently published (MitoCarta3.0, Rath et al, 2021), the authors could perform a GSEA using MitoPathways as signature, thereby focusing on mitochondrial changes occurring exercise training. Considering the focus of the present study, this might be more informative than a general GO enrichment analysis.

Is it possible to apply the same kind of normalization to the transcriptome analysis? E.g. filtering read counts from high-confidence IMPI and using this to normalize the data from mitochondrial-related

genes. This would be informative and could turn out to be a valuable tool considering the number of pre-existing RNA-seq datasets following exercise training.

As a proof-of-concept, it would be interesting to apply the same normalization approach to publicly available MS data from different muscle fiber types (e.g. Murgia et al, Cell Reports, 2017).

It is dangerous to draw very strong conclusions from the fact that transcriptomics data don't match or explain proteomics data. There are several layers of regulation in between gene transcription and protein levels and activity, which could explain this apparent disconnect. mRNA stability, increased protein translation, import into mitochondria, protein stability, and post-translational modifications.

In addition, one often neglected aspect is alternative splicing, which is also not easily identifiable in standard RNA-sequencing analysis. It is, however, increasingly appreciated that upon stimulation cells can change the splicing of transcripts already expressed in that cell, to achieve rapid changes in protein structure/function. These changes in splicing often don't show up in RNA-seq analysis, due to the standard way in which algorithms normalize RNA-seq reads by gene "size" or annotated exon structure.

There is also a quantitative aspect to consider. Many of these gene expression changes are quite small, and also often fall out of the statistical analysis – especially when there is large variability between samples, which is to be expected in human studies. On the other hand, a 10% increase in mitochondrial gene / protein expression might be quite significant.

Reviewer #3 (Remarks to the Author):

The authors perform an impressive study involving 4 consecutive muscle biopsies, a design that is difficult to execute in most places in the world. They combine state-of-the art transcriptomics, proteomics, and lipidomics to address an important question around the exercise-induced biological changes within mitochondria. The findings are well illustrated and discussed within an exceptionally well-delineated review of the literature. The paper is well written, relatively concise, and insightful, making significant new observations related to the timing and to the nature of exercise-induced molecular adaptations in human skeletal muscle mitochondria. Below I make some suggestions to help improve the manuscript.

1. Could the apparent "deprioritization" of ETC proteins be the result of their degradation in the early phase of the adaptation to exercise? Have the authors investigated mitochondrial proteases, proteins

related to mitochondrial derived vesicle (MDV) formation, or other quality control or turnover processes?

2. Two studies have reported a decrease in mitochondrial DNA (mtDNA) content following exercise (see below), despite preserved mitochondrial energetics. Have the authors examined proteins related to mtDNA maintenance and replication? Or mtDNA copy number directly from the biopsy material. Do changes follow the trajectory of ETC complexes, or of other mitochondrial components? Adding this information to Figure 3, or to a new figure along with mitochondrial turnover could be informative, and could help further interpret the potential origin of the robust decrease in ETC proteins after two weeks of exercise.

Egan B, O'Connor PL, Zierath JR, O'Gorman DJ (2013) Time course analysis reveals gene-specific transcript and protein kinetics of adaptation to short-term aerobic exercise training in human skeletal muscle. *PLoS One* 8: e74098. doi: 10.1371/journal.pone.0074098

Puente-Maestu L, Lazaro A, Tejedor A, Camano S, Fuentes M, Cuervo M, Navarro BO, Agusti A (2011) Effects of exercise on mitochondrial DNA content in skeletal muscle of patients with COPD. *Thorax* 66: 121-7. doi: 10.1136/thx.2010.153031

To measure mtDNA levels, two approaches could be taken. The first would be to quantify mtDNA copy number in the standard way from whole muscle homogenate (mtDNA:nDNA ratio). But this would be confounded by overall changes in mitochondrial content. The second and perhaps more informative approach consistent with the author's approach would be to quantify mtDNA abundance in the isolated mitochondria (by digital or quantitative PCR), and to report absolute or relative mtDNA levels normalized to mitochondrial proteins, or to cardiolipins. This would reveal if mtDNA abundance on a per-mitochondrion basis changes with exercise, and add valuable information about the mitochondrial genome adaptations in the context of the comprehensive transcriptional, proteomic, and lipidomic adaptations.

3. Figure 4: the abbreviations for each lipid species in panel (d) should be defined in the figure legend.

4. The title of figure 5 should be changed to avoid interpretation, and rather refer to the content of the figure.

AUTHOR RESPONSE TO REVIEWER COMMENTS

Reviewer #1 (Remarks to the Author):

The authors performed multi-omic analyses in a well-controlled study of human exercise training. The authors should be commended for this. They undertake a rigorous analysis of the mitochondrial proteome, including a simple but effect means of normalizing the data for exercise-induced mitochondrial biogenesis. These are really impressive data and I recommend that this manuscript should be published with only minor modifications:

RESPONSE: We really appreciate the comments and the time invested reviewing our work, and would like to thank the Reviewer for helping us to improve the manuscript. All changes made have been tracked in the manuscript to assist the Reviewers; line numbers in this document refers to the “tracked changes” manuscript version.

Supplemental Fig 1: Please show representative images of immunoblots

RESPONSE: These are now presented in a separate file entitled “CGranata_uncropped BN- and SDS-PAGE images”, as per Nature Communications Editorial Policy; these include uncropped images used to prepare Fig. 5a, and uncropped images from SDS-PAGE and BN-PAGE blots used in quantification of data presented in Supplementary Fig. 1a, Supplementary Fig. 2b, and Fig. 5b.

Fig 1F: Colour could be incorporated to better highlight the profiles of the selected proteins, as in 2C and 4D.

RESPONSE: As suggested by the reviewer, we have changed the colours used in the profile plots in Fig. 1f to better highlight the profiles of the selected transcripts, as per other figures in this manuscript.

Fig 1G: Are the transcriptomics enrichment analyses performed as an “average” of the three post-training timepoints? Please describe this method in further detail (in the methods section).

RESPONSE: Following the Reviewer’s request, we now more clearly explain this in the Figure legends (lines 160 to 162) and the relevant Methods section (lines 852 to 868). In summary, similar to what we did for the other omics techniques, we calculated pairwise comparisons of transcript read counts for each sequential training volume comparison (i.e., PN vs. BL, PH vs. PN, and PR vs. PH). Linear modelling in *limma* (comparable to a multiple samples ANOVA) was performed on the three comparisons to identify transcripts that were differently altered across the four training phases. These transcripts and their changes are visualised in the heatmap presented in Fig. 1e. An enrichment analysis, using Enrichr, was then performed on the same list of differentially altered transcripts with the results shown in Fig. 1g and a in a new tab (entitled “GOBP-Enrichr”) added to Supplementary Table 2.

Lines 130-133: When normalized to CS activity many of the increases in respiratory states and complex activity, which are apparent as absolute values, are abolished in the PH condition. Please highlight and discuss this in text in relation to effects of training duration, volume and intensity on mitochondrial content and efficiency. At least refer to similar observations you made in Granata et al (2016) (<https://pubmed.ncbi.nlm.nih.gov/27402675/>).

RESPONSE: Following the Reviewer's request we now discuss these findings in relation to previous literature in lines 134 to 138 of the revised manuscript.

Line 166: Whilst I appreciate the validity and utility of the transcriptomic data, transcriptomic data cannot provide an insight into protein stoichiometry. Please amend the sentence on lines 166-167.

RESPONSE: It was not our intention to suggest a link between transcriptomics and proteomics findings. We agree with the Reviewer that as written this may have been misinterpreted this way, and we have now amended the sentence as requested. The new sentence can be found on lines 173 to 174 and reads "To gain further insight into the effects of training on mitochondria we first employed RNA sequencing (RNA-seq) based transcriptomics".

Line 175: Only the top 5 enriched GOBP terms are displayed in Fig 1g, therefore it is impossible for readers to verify that mitochondrial terms were not enriched. Please display all of the enrichment terms, for example in a bubble plot, or, if there are too many to display, please refer to a supplemental table.

RESPONSE: Due to a recent update of Enrichr from the 2018 GOBP database to the 2021 GOBP database, we had to regenerate the enrichment calculations and therefore reproduce Fig. 1g. The general trend remains similar and still highlights that mitochondrial terms were not significantly enriched following exercise training. All identified terms and their associated scores can now be found in a new tab (entitled "GOBP-Enrichr") added to Supplementary Table 2; as a consequence of the update to the GOBP database, the figure legend of Fig. 1g (lines 160 to 162) and the relevant paragraph in the Methods section (lines 852 to 868) have also been updated.

Figs 2E, 3 & 4C: Whilst I can interpret the heat maps shown in these figures, in which the fold change of each training block is compared to the previous, I would find it easier to follow the time-course of the adaptations if each training blocks was compared to baseline.

RESPONSE: We agree that comparing each time point with baseline provides valuable information. As such, this information was already available in our Figures and Supplementary materials. Supplementary Table 7 shows the list of upregulated and downregulated proteins for all 6 time point comparisons, including PN vs. BL, PH vs. BL, and PR vs. BL, as suggested; Fig. 2g represents these comparisons visually. Similarly, the forest plots in Supplementary Figure 3 show all upregulated and downregulated lipid species for all 6 time point comparisons, including PN vs. BL, PH vs. BL, and PR vs. BL, as suggested by the Reviewer. Six new tabs have now been added to Supplementary Table 9 detailing these comparisons. While we have opted not to include heatmaps showing

comparisons of the 3 post-training time points vs. BL to avoid potential confusion, we have now added a sentence in each relevant section more clearly pointing the reader toward these resources (lines 331 to 333 and 443 to 445). Finally, we would like to point out that we used sequential comparisons so that the specific changes induced by each training phase could be clearly identified. This reflects the experimental design, in that phases were run sequentially, and there was no return to baseline after each phase. In line with this, we felt it was more appropriate to present results originating from comparisons between sequential time points.

Pages 27, 28 and 29: Reference should be made to the (very) newly published study from Gonzalez-Franquesa et al (2021) (<https://pubmed.ncbi.nlm.nih.gov/34038727/>), in which SC formation and COQ protein complex formation was studied in muscle in the context of exercise training, albeit in mice.

RESPONSE: We now cite the paper in question, which was not yet published at the time of our paper submission, both in the context of mitochondrial complex re-distribution within supercomplexes following exercise training (lines 545 to 546) and CoQ adaptation (lines 580 to 581).

Results & Discussion: I find the order of the results is illogical – specifically it seems strange to jump from proteomics (fig 3) to lipidomics (fig 4) and back again. In my opinion, the results and discussion pertaining to COQ biosynthesis and cristae formation should also come earlier.

RESPONSE: As with most publications, we agree there is more than one way that the results could have been presented. The reasoning behind the chosen order is that proteomics constituted the main technique used, making this a proteomics study in our view. As a consequence, lipidomics, as well as other techniques, were used as valuable tools to introduce, support, integrate, and expand the findings provided by the proteomics assessment. As to the position of CoQ and cristae formation in the text, we agree these could have also been presented earlier, but for simplicity we decided to present the results based on the protein clusters identified in our proteomics enrichment analysis. In this regard, we feel that following the discussion of findings related to changes in OXPHOS subunits, it made sense to discuss the results of our BN-PAGE assessment of supercomplexes, before moving on to discussing the next clusters (CoQ and cristae formation).

Lines 626-627: It should be considered that some of the changes seen may be due to timing (i.e. duration of the training programme) rather than training volume – after all two weeks is a relatively short duration and some exercise-induced adaptations may take longer. For example, translatory proteins may be one of the early adaptations to exercise training/an increase in volume, whilst the expression of OXPHOS complex proteins may follow.

RESPONSE: We agree with this observation and we have modified the sentence originally at line 626-627 to reflect this. It now reads (lines 658) “Although the influence of training duration cannot be completely ruled out, the above findings were the result of changes in only one of the several exercise prescription programming variables (i.e., training volume)”. We added a similar statement in the “protein translation” section stating this caveat (lines 356 to 357).

Reviewer #2 (Remarks to the Author):

The manuscript by Granata et al. uses a combination of biochemical, transcriptomic, proteomic and lipidomic approaches to evaluate changes on skeletal muscle mitochondria after a period of exercise training in humans. To distinguish the effects of overall increased mitochondrial content after exercise training from more specific mitochondrial adaptations, the authors apply an in silico normalization to both proteomics and lipidomics data. This allowed the identification of protein/lipid clusters that the authors propose to be differentially prioritized during exercise training. After observing that changes in mitochondrial constituents are not stoichiometrically linked to the overall mitochondrial content, the authors show that TCA cycle and FAO are prioritized over the expression of OXPHOS subunits. This, together with the data on mitochondrial supercomplexes, led them to conclude that during exercise training enhancing electron flow to OXPHOS is more important for ATP generation than the overall OXPHOS content.

The study is very well designed and represents a clear sophistication of the group's research efforts, which are very well known in the field of exercise physiology. The mitochondrial phenotyping is impressive, especially considering the logistics involved in collecting the samples and the limited amount of material to work with.

Conceptually, the study addresses a long-standing problem of how to decouple mitochondrial content from respiratory function. Although the data herein presented constitute an important step towards solving this issue, the conclusions put forward in the manuscript are quite strong (sometimes provocative) and not entirely supported by the data, remaining mostly associative and descriptive. Indeed, that might be the main shortcoming of the work, as it sets the frame for the identification of novel players in mitochondrial function, but then it concludes without really advancing our understanding of how skeletal muscle mitochondria adapt to exercise training from a mechanistic perspective. This is emphasized by the fact that previous research, including work from the same group, has already shown some disconnect between mitochondrial content and function (Granata et al 2016), highlighting the importance of TCA cycle and FAO for early adaptations to exercise training.

RESPONSE: We appreciate the comments and the time invested reviewing our work, and would like to thank the Reviewer for helping us to improve the manuscript. In response to the Reviewer's suggestion that some of the conclusions in this manuscript could be considered quite strong, we have toned down the stronger assertions made throughout the manuscript. As noted by this reviewer, one of the strengths of this work is as a resource paper and we have tried to better emphasise this aspect in the revised manuscript. All changes made have been tracked in the manuscript to assist the Reviewers; line numbers in this document refers to the "tracked changes" manuscript version.

Here are some points that could improve the general impact and message of the study. I don't think it's fair to call this a multi-omics analysis (rather a "multiple-omics") as the different omics data are not analyzed in a truly integrated way. This would require some additional (and not trivial) bioinformatics analysis, where all the different layers of data are co-analyzed to identify connections and correlations between them. This would definitely improve the study's impact, but this type of analyses are a project in itself.

RESPONSE: We agree with the Reviewer that to be defined as a true "multi-omics" study, the manuscript would have likely necessitated an integrated analysis of the different omics

techniques. We have now changed “multi-omics” to “multiple omics” throughout the manuscript (lines 35, 90, and 624).

Contrary to what the authors had observed before (Granata 2016), a reduction of markers for mitochondrial content post-RVT was not seen. This could be explained by a shorter RVT period and/or increased volume compared with the previous study (6 sessions over 1 week vs 5 sessions over 2 weeks). The authors should comment on how this could affect the results herein presented, especially considering the mitochondrial proteome rearrangement. Would it be expected that OXPHOS proteins decrease (prioritized removal) faster than TCA- and FAO-related proteins?

RESPONSE: Thank you for this comment, and we agree that the different findings can most likely be attributed to the shorter RVT period and/or increased volume compared with our previous study. We have now made a comment to put the current manuscript results in context with the previous findings from Granata et al. 2016 (see lines 339 to 343). We agree that it would be very interesting to examine changes in the mitochondrial proteome in response to a longer RVT period involving a greater reduction in training volume but we prefer not to speculate beyond our data in the current manuscript.

Since in essence this is a resource paper, the transcriptome analysis could have been better explored, e.g., comparing the changes observed in the mitochondrial proteome with corresponding transcripts. Is there a clear disconnect between transcripts and proteins? A heatmap showing the transcriptional changes of the same genes presented in Fig 2e could be helpful (highlighting those that achieved significance). Taking advantage of the MitoPathways annotations recently published (MitoCarta3.0, Rath et al, 2021), the authors could perform a GSEA using MitoPathways as signature, thereby focusing on mitochondrial changes occurring exercise training. Considering the focus of the present study, this might be more informative than a general GO enrichment analysis.

RESPONSE: We agree that it can be valuable to compare changes observed in the transcriptomics and proteomics analyses. However, performing this analysis with our current datasets would be inappropriate, as transcriptomics was performed on whole-cell extracts, whereas proteomics was performed on mitochondrial isolates. The large and significant increase in mitochondrial protein enrichment (MPE) throughout the training intervention (Fig. 2a) would likely complicate any attempts at a truly integrated multi-omic analysis. Another complication stems from the fact that due to lack of available material transcriptomics was only performed on 5 participants (whereas proteomics and lipidomics was performed on all 10 participants). Even if a true multi-omics analysis was possible using our dataset the Reviewer rightly states “*it is dangerous to draw very strong conclusions from the fact that transcriptomics data don’t match or explain proteomics data*”, which is in line with debate in the field on the validity of the correlation between global changes in mRNA and protein changes following a training intervention (Miller et al. 2016 [PMID: 27013604]; Hornberger et al. 2016 [27543661]; Miller et al. 2016 [PMID: 27543662]), and previous literature showing a generally poor correlation between changes in protein and mRNA following a perturbation (Vogel et al. 2011 [PMID: 21933953]; Fournier et al. 2010 [PMID: 19955083]; De Godoy et al. 2008 [PMID: 18820680], Lee et al. 2011 [PMID: 21772262];

Jayapal et al. 2008 [PMID: 18461186]; Maier et al. 2011 [PMID: 21772259]; Kristensen et al. 2013 [PMID: 24045637]).

Is it possible to apply the same kind of normalization to the transcriptome analysis? E.g. filtering read counts from high-confidence IMPI and using this to normalize the data from mitochondrial-related genes. This would be informative and could turn out to be a valuable tool considering the number of pre-existing RNA-seq datasets following exercise training.

As a proof-of-concept, it would be interesting to apply the same normalization approach to publicly available MS data from different muscle fiber types (e.g. Murgia et al, Cell Reports, 2017).

RESPONSE: This is an interesting proposal; however, we believe it would not be appropriate to use a similar normalization approach to normalize our transcriptome analysis. The principal reason is that our proteomics assessment required a normalization approach to eliminate the bias introduced by the large and significant increase in mitochondrial content post-training. This increase would not occur at the transcript level since transcripts encoding mitochondrial proteins are located in the cytosol and get eventually degraded, contrary to the fate of mitochondrial proteins that accumulate and result in an increase in mitochondria content. An exception would be the 12 mtDNA derived mRNAs; however, they were not included in our transcriptome assembly due to technical limitations. In respect to the second part of this comment, regarding benchmarking our normalization approach on another dataset, this is very interesting and it is something we are exploring for another manuscript currently under preparation. We believe this type of benchmarking analysis remains outside of the scope of this paper.

It is dangerous to draw very strong conclusions from the fact that transcriptomics data don't match or explain proteomics data. There are several layers of regulation in between gene transcription and protein levels and activity, which could explain this apparent disconnect. mRNA stability, increased protein translation, import into mitochondria, protein stability, and post-translational modifications.

RESPONSE: We believe the Reviewer's comment refers to the sentence originally at line 166-167, as also remarked by Reviewer 1. It was not our intention to draw any direct correlation between transcriptomics and proteomics; to avoid any confusion and misinterpretation, we have now reworded that sentence to read "To gain further insight into the effects of training on mitochondria we first employed RNA sequencing (RNA-seq) based transcriptomics" (see lines 173 to 174).

In addition, one often neglected aspect is alternative splicing, which is also not easily identifiable in standard RNA-sequencing analysis. It is, however, increasingly appreciated that upon stimulation cells can change the splicing of transcripts already expressed in that cell, to achieve rapid changes in protein structure/function. These changes in splicing often don't show up in RNA-seq analysis, due to the standard way in which algorithms normalize RNA-seq reads by gene "size" or annotated exon structure.

There is also a quantitative aspect to consider. Many of these gene expression changes are

quite small, and also often fall out of the statistical analysis – especially when there is large variability between samples, which is to be expected in human studies. On the other hand, a 10% increase in mitochondrial gene / protein expression might be quite significant.

RESPONSE: We agree that an analysis of alternative splicing would be very novel and informative. However, the aim of this study was to characterize the mitochondrial proteome following different training volumes; as such, transcriptomics, as well as lipidomics and other classic biological techniques, were mainly used to provide support for observations at the protein level. Further, due to limitations in sample availability, we were able to perform transcriptomics only on a reduced sample size ($n = 5$). As a consequence, the validity of the information obtained by investigating alternative splicing would be considerably hampered. Taken together, we believe that an analysis of alternative splicing remains outside of the scope of this manuscript.

Reviewer #3 (Remarks to the Author):

The authors perform an impressive study involving 4 consecutive muscle biopsies, a design that is difficult to execute in most places in the world. They combine state-of-the-art transcriptomics, proteomics, and lipidomics to address an important question around the exercise-induced biological changes within mitochondria. The findings are well illustrated and discussed within an exceptionally well-delineated review of the literature. The paper is well written, relatively concise, and insightful, making significant new observations related to the timing and to the nature of exercise-induced molecular adaptations in human skeletal muscle mitochondria. Below I make some suggestions to help improve the manuscript.

RESPONSE: We really appreciate the comments and the time invested reviewing our work, and would like to thank the Reviewer for helping us to improve the manuscript. All changes made have been tracked in the manuscript to assist the Reviewers; line numbers in this document refers to the “tracked changes” manuscript version.

- 1.** Could the apparent “deprioritization” of ETC proteins be the result of their degradation in the early phase of the adaptation to exercise? Have the authors investigated mitochondrial proteases, proteins related to mitochondrial derived vesicle (MDV) formation, or other quality control or turnover processes?

RESPONSE: This is a very valid suggestion by the Reviewer. We originally made a similar hypothesis and performed *ad hoc* searches on proteins involved in quality control processes or protease degradation. Despite being able to identify several of these proteins (AFG3L2, CLPP, CLPX, HTRA2, LONP1, PITRM1), only CLPP was significantly altered following training (increased PN vs. BL; Supplementary Table 7). Although it is tempting to speculate that the deprioritization of OXPHOS subunits post-NVT could stem from an increase in proteolytic activity induced by CLPP, we refrained from including this observation as it would be reasonable to expect its partner protein CLPX, as well as proteases known to degrade OXPHOS subunits (particularly LONP1 and AFG3L2) to be similarly altered.

- 2.** Two studies have reported a decrease in mitochondrial DNA (mtDNA) content following exercise (see below), despite preserved mitochondrial energetics. Have the authors examined proteins related to mtDNA maintenance and replication? Or mtDNA copy number directly from the biopsy material. Do changes follow the trajectory of ETC complexes, or of other mitochondrial components? Adding this information to Figure 3, or to a new figure along with mitochondrial turnover could be informative, and could help further interpret the potential origin of the robust decrease in ETC proteins after two weeks of exercise.
 - a.** Egan B, O'Connor PL, Zierath JR, O'Gorman DJ (2013) Time course analysis reveals gene-specific transcript and protein kinetics of adaptation to short-term aerobic exercise training in human skeletal muscle. PLoS One 8: e74098. doi: 10.1371/journal.pone.0074098
Puente-Maestu L, Lazaro A, Tejedor A, Camano S, Fuentes M, Cuervo M, Navarro BO, Agusti A (2011) Effects of exercise on mitochondrial DNA content in skeletal muscle of patients with COPD. Thorax 66: 121-7. doi: 10.1136/thx.2010.153031

- b. To measure mtDNA levels, two approaches could be taken. The first would be to quantify mtDNA copy number in the standard way from whole muscle homogenate (mtDNA:nDNA ratio). But this would be confounded by overall changes in mitochondrial content. The second and perhaps more informative approach consistent with the author's approach would be to quantify mtDNA abundance in the isolated mitochondria (by digital or quantitative PCR), and to report absolute or relative mtDNA levels normalized to mitochondrial proteins, or to cardiolipins. This would reveal if mtDNA abundance on a per-mitochondrion basis changes with exercise, and add valuable information about the mitochondrial genome adaptations in the context of the comprehensive transcriptional, proteomic, and lipidomic adaptations.

RESPONSE: We appreciate the suggestion by the Reviewer. Similar to the previous discussion regarding the investigation of proteases and quality control processes, our discussion and analysis was guided by the results of our pathway enrichment analysis, which did not highlight changes in processes related to mtDNA maintenance and replication. Following the Reviewer's suggestion, we further inspected our proteomics results and performed *ad hoc* searches, but we did not identify any significant change to proteins related to mtDNA maintenance. In addition, the two suggested references show a decrease in mtDNA only in the early recovery phase following a single session of exercise, suggesting this may be only an acute adaptation. On the contrary, the Egan et al. 2013 [PMID: 24069271] study reported no change in mtDNA following a 14-session training intervention. Lastly, and more importantly, a seminal study from Larsen et al 2012 [PMID: 22586215] demonstrated that mtDNA is not a strong predictor of mitochondrial content, as it shows only a modest correlation to it ($r = 0.35$); this is also one of the reasons why we used different markers of mitochondrial content in the current study. For these reasons, and due to the lack of further biopsy material, we are not in a position to perform this analysis, nor to speculate on this interesting topic.

3. Figure 4: the abbreviations for each lipid species in panel (d) should be defined in the figure legend.

RESPONSE: We thank the Reviewer for having spotted this omission; all abbreviations for each lipid species in Fig. 4d have now been defined in the figure legend (lines 416 to 426).

4. The title of figure 5 should be changed to avoid interpretation, and rather refer to the content of the figure.

RESPONSE: The title of figure 5 has been changed, as suggested. It now reads "Training-induced changes in mitochondrial content, but not changes in the abundance and/or organisation of SCs, occur alongside improvements in mitochondrial respiration" (lines 519 to 521).

REVIEWERS' COMMENTS

Reviewer #1 (Remarks to the Author):

The author have done excellent job in answering all my concerns. We do not have any further comments.

Reviewer #2 (Remarks to the Author):

Although in reality the authors didn't really change the manuscript much in line with my suggestions and recommendations, I do think that the paper stands as a resource paper as it is. Rephrasing the stronger statements has improved the manuscript.

Reviewer #3 (Remarks to the Author):

All comments have been addressed

AUTHOR RESPONSE TO REVIEWER COMMENTS

Reviewer #1 (Remarks to the Author):

The author have done excellent job in answering all my concerns. We do not have any further comments.

RESPONSE: We thank the Reviewer for the kind comments and the time invested reviewing our work, which has helped improve the manuscript.

Reviewer #2 (Remarks to the Author):

Although in reality the authors didn't really change the manuscript much in line with my suggestions and recommendations, I do think that the paper stands as a resource paper as it is. Rephrasing the stronger statements has improved the manuscript.

RESPONSE: We thank the Reviewer for the kind comments and the time invested reviewing our work, which has helped improve the manuscript.

Reviewer #3 (Remarks to the Author):

All comments have been addressed

RESPONSE: We thank the Reviewer for the kind comments and the time invested reviewing our work, which has helped improve the manuscript.